# LeIn-PINN: Learned Initialization to Alleviate Convergence Failures in Physics Informed Neural Networks

## Abstract

Physics-informed neural networks (PINN) have had a broad research impact in modeling domains governed by partial differential equations (PDE). However, PINNs have been shown to perform sub-optimally or even converge to trivial solutions, in challenging scenarios, such as stiff PDE domains, or when generalizing to unseen but related experimental contexts. Previously proposed solutions to alleviate catastrophic PINN failures, detail curriculum learning based training techniques or training techniques involving dynamic re-sampling of *hard* collocation points. We observe that these methods face certain pitfalls e.g., designing a training curriculum is ambiguous in multi-parameter PDE settings and dynamic resampling of collocation points still fails in complex PDE settings. Recent works have highlighted that conflicting gradients during PINN training, may be one of the major influences of such catastrophic PINN failures. Complementary to this line of thinking, we believe the initialization of PINN weights also plays a crucial role in the emergence of catastrophic failures during training. To this end, we propose a novel PINN training methodology based on *Learned Initialization* (LeIn) to address catastrophic PINN failures. We call our variant LeIn-PINN. Through rigorous experiments on 1D and 2D PDE domains (including evaluation in challenging 2D fluid dynamics contexts), we demonstrate that our proposed methodology outperforms state-of-the-art methods, including ones specifically designed to alleviate PINN failures. We show that LeIn-PINNs achieve an average performance improvement of **89%** over state-of-the-art baselines. Further, we also conduct a rigorous analysis to identify and explain the improved training dynamics of LeIn-PINNs as well as the convergence failure of traditional PINNs, by analyzing the characteristics of loss landscape of the respective converged models. Finally, we also demonstrate that our proposed LeIn-PINN method significantly reduces spectral bias compared to traditional PINNs even in challenging PDE domains.

## 1 Introduction

Partial differential equations (PDEs) govern dynamics in heat transfer, fluid flow, electromagnetics, and other physical systems. Efficient PDE solvers are central to scientific progress. Physics-Informed Neural Networks (PINNs) (Raissi et al., 2017b; 2018) embed PDE constraints into neural network loss functions, enabling both forward and inverse problem solving. PINNs have been applied widely (Cuomo et al., 2022b), but often fail in *hard* PDE settings, converging to trivial or inaccurate solutions. A key challenge is the imbalance between data-driven and physics-informed loss terms (Wang et al., 2020), which causes optimization pathologies. Remedies include curriculum learning (Krishnapriyan et al., 2021), adaptive sampling (Daw et al., 2023), and spectral/Neural Tangent Kernal(NTK) analyses (Wang et al., 2021b; 2022b; Lau et al., 2024). Yet these approaches typically succeed only in narrow PDE regimes and lack generality. Meanwhile, the ML community has shown that initialization strongly influences trainability in deep networks (Arpit et al., 2019; Schoenholz et al., 2017; Taki, 2017; Yang & Schoenholz, 2017). Standard schemes such as Xavier or Kaiming can place networks in chaotic or overly ordered regimes, leading to poor convergence. Despite this, nearly all PINN studies still rely on off-the-shelf initializers without task-specific adaptation.

To address this gap, we introduce **LeIn-PINN**, a learned initialization framework for PINNs that alleviates catastrophic training dynamics. Our contributions are as follows:

- Ours is the first and (to the best of our knowledge) only work to investigate and propose a systematic, intelligent initialization strategy to alleviate catastrophic failures, prevalent in Physics-Informed Neural Networks (PINN) while modeling challenging PDE domains.

- We also introduce a novel gated layer-wise optimization (GLO) procedure and demonstrate that it mitigates spectral bias in the PINN training context.

- Our extensive experimental evaluations on three distinct PDE domains, namely: 1D-Convection, 2D-Helmholtz, and 2D-Navier-Stokes, demonstrate consistent and significant gains over multiple state-of-the-art baselines in challenging extrapolation regimes.

## 2 RELATED WORK

A broad set of research has addressed PINN training failures (Cuomo et al., 2022a). Here we highlight the most recent, representative works in each research direction.

**Temporal Dynamics, Sampling, and Curriculum-Based Methods.** A seminal work (Wang et al., 2021a) uncovered the imbalance in gradients between data-driven and physics-based loss terms as one of the reasons for catastrophic PINN failures, causing converge to trivial solutions. Early work also highlighted the importance of respecting causal order of PDE events during training, improving convergence in time-dependent problems (Wang et al., 2022a). Building on this, residual-based R3 sampling techniques dynamically reweigh collocation points in regions with high PDE residuals (Daw et al., 2023) (Toloubidokhti et al., 2023). In parallel, curriculum learning approaches schedule training tasks or progressively increase collocation difficulty, guiding PINNs from simpler to complex PDE regimes (Krishnapriyan et al., 2021). While these strategies reduce domain errors, they require careful tuning when PDE parameters vary widely and do not generalize well across different systems, task difficulties.

**Gradient-Level Interventions** Efforts to address the issue of loss imbalance in PINNs have also explored gradient-level solutions. For instance, Kim et al. (2021) introduced a Dynamic Pulling Method (DPM) which employs a pseudo-inverse operation to align losses in the same direction, thereby improving stability, effectiveness of training. Wang et al. (2025) has explored the effect of higher order optimization techniques.

**Meta-Learning for PINNs.** Several recent works have explored meta learning concepts in PDE systems. Psaros et al. (Psaros et al., 2022) uses meta learning to discover optimal loss function for different PINN system. In contrast LeIn-PINN uses meta learning to get a learned initialization. Hyper-LR-PINNs (Cho et al., 2023) employ a hyper-network architecture to generate low-rank, task-specific weights for each PDE domain. Although they have trained on similar PDE domains to ours, their experiments have not explored extrapolation regimes as extreme as ours in this paper. Qin et al. (2022) is the closest paper to ours and also employs a standard MAML (Model Agnostic Meta Learning) approach for PINN training. However, their evaluation is restricted to relatively simpler PDE configurations compared to our evaluation domains and they do not explore extrapolation ability of the proposed model.

In contrast to other approaches to alleviate catastrophic training failures in PINN, LeIn-PINN is the only approach to systematically explore the effect of learned weight initialization on PINN training dynamics. Further, our gated layer-wise optimization is the first approach of its kind applied in the context of addressing convergence failures in PINNs across diverse PDE settings.

## 3 PROBLEM FORMULATION

We follow the standard PINN setting (Raissi et al., 2017a). We provide a brief primer here; the PDE system specific detailed formulation is in Appendix B. Let $\mathbf{z} \in \Omega \subset \mathbb{R}^d$ denote the spatial coordinate within the domain $\Omega$ (with boundary $\partial\Omega$), $t \in [0, T]$ time, and $u(\mathbf{z}, t)$ the physical state solving the PDE in Eq. 1.

$$\mathcal{F}\big(u(\mathbf{z}, t); \gamma\big) = f(\mathbf{z}, t), \quad \mathbf{z} \in \Omega, \ t \in [0, T], \qquad \mathcal{B}\big(u(\mathbf{z}, t)\big) = u_b(\mathbf{z}, t), \ \mathbf{z} \in \partial\Omega, \qquad (1)$$

Here, $\mathcal{F}$ is the (possibly nonlinear) differential operator representing the underlying physical law, $\gamma$ are the PDE parameters (e.g., viscosity, diffusivity), and $f(\mathbf{z}, t)$ is a known forcing or source term.

The boundary operator $\mathcal{B}$ enforces boundary or initial conditions, and $u_b(\mathbf{z}, t)$ specifies the prescribed values along $\partial\Omega$ and the subscript '$b$' is added to indicate solution along the boundary $\partial\Omega$.

A neural network $\hat{u}_\Theta(\mathbf{z}, t)$ parameterized by weights $\Theta$ is trained to satisfy the PDE at interior (collocation) points and to match boundary/initial data on $\partial\Omega$. With $N_\Omega$ interior samples $(\mathbf{z}_i, t_i)$ and $N_{\partial\Omega}$ boundary/initial samples $(\mathbf{z}_j, t_j)$, the objective is listed in Eq. 2.

$$\mathcal{L}(\Theta) = \lambda_r \frac{1}{N_\Omega} \sum_{i=1}^{N_\Omega} \left| \mathcal{F}(\hat{u}_\Theta(\mathbf{z}_i, t_i); \gamma) - f(\mathbf{z}_i, t_i) \right|^2 + \frac{1}{N_{\partial\Omega}} \sum_{j=1}^{N_{\partial\Omega}} \left| \hat{u}_\Theta(\mathbf{z}_j, t_j) - u_b(\mathbf{z}_j, t_j) \right|^2. \quad (2)$$

Here, $\lambda_r$ balances the contributions of the physics residual and boundary losses.

We study three canonical PDE systems commonly used to benchmark scientific machine learning: 1D Convection, the 2D Helmholtz equation, and the 2D incompressible Navier–Stokes equations. Each captures a distinct physical regime—transport, wave propagation, and nonlinear fluid dynamics—that poses different challenges for PINN optimization.

Although the PINN formulation has been highly impactful in solving *forward* problems across variegated PDE domains, it has recently been found that PINNs fail to learn faithful solutions (and converge to trivial solutions) in challenging (e.g., stiff) PDE contexts. Previous work (Wang et al., 2020) has characterized the conflicting and imbalanced gradients between the two loss terms in $\mathcal{L}(\Theta)$ as one of the reasons for such failures. Recent approaches have proposed solutions to address PINN convergence failures by R3 re-sampling (Daw et al., 2023; Wu et al., 2023), curriculum based PINN training (Krishnapriyan et al., 2021).

Complementary to these works, we hypothesize that PINN convergence failures in *challenging* (a.k.a. hard) PDE domains, can be alleviated, if the weights $\Theta$ of the PINN, instead of being randomly initialized, are systematically initialized prior to training on the *hard* domain. To this end, we propose a *Learned-Initialization* (LeIn) mechanism to systematically learn better initial weights, such that PINNs trained with LeIn weights overcome catastrophic failures in challenging PDE domains. Our proposed LeIn approach has two facets (i) Invariance Encoding (ii) Gated Layer-wise Optimization.

**Invariance Encoding (IE).** If the goal of the PINN is to learn a challenging PDE domain governed by PDE parameter $\gamma_{\text{hard}}$, IE seeks to first train a randomly initialized PINN model on a set of *relatively easier* configurations of the PDE with parameters $\Gamma_{\text{easy}} = \{\gamma_1, \ldots, \gamma_K\}$. This enables a distillation of the *invariant* physics across $\Gamma_{\text{easy}}$ into the PINN weights $\Theta$. Our approach for IE is grounded in foundational works of meta-learning (Finn et al., 2017). IE begins with randomly initialized *global* PINN weights $\Theta_0$ and pool of $K$ tasks in $\Gamma_{\text{easy}}$. The IE learning process is carried out for $\mathcal{R}$ iterations $\{1, 2, \ldots, \mathcal{R}\}$ and at each training iteration $r$, tasks $\{\mathcal{T}_1, \ldots, \mathcal{T}_k\}$ are randomly sampled from the task distribution $p(\Gamma_{\text{easy}})$ of easy tasks and $k$ identical copies $\{\Theta'_{r,1}, \ldots, \Theta'_{r,k}\}$ of the current global PINN weights $\Theta_r$ are created. Each $\Theta'_{r,i}$ is then optimized via. gradient descent, employing data from task $\mathcal{T}_i$ at iteration $r$ for 'J' *inner* iterations (for simplicity we assume 'J' = 1). Let $\Theta''_{r,i}$ represent the updated version of $\Theta'_{r,i}$ after 'J' inner gradient-descent update iterations. The *common* representations across all $k$ task models $\{\Theta''_{r,1}, \ldots, \Theta''_{r,k}\}$ are distilled into $\Theta_r$ via. a 'meta-update' as indicated in Eq. 3 where $\mathcal{L}_{\mathcal{T}_i}(\Theta''_{r,i})$ represents the loss in Eq. 2 calculated w.r.t parameters $\Theta''_{r,i}$ with data from task $\mathcal{T}_i$ and $\eta_{\text{outer}}$ represents the learning rate.

---

**Algorithm 1:** Physics Informed Neural Network with Learned Initializations

**Input:** Tasks $p(\Gamma_{\text{easy}})$, initial weights $\Theta_0$, inner LR $\eta_{\text{inner}}$, outer LR $\eta_{\text{outer}}$, total meta-iterations $\mathcal{R}$, Num. Sampled Tasks $k$

**Output:** LeIn Weights $\Theta_\mathcal{R}$

**for** $r \leftarrow 0$ **do** // meta-iterations
  $r = 1, \ldots, \mathcal{R}$

3     Sample batch $\{\mathcal{T}_i\}_{i=1}^k \sim p(\Gamma_{\text{easy}})$

5     Copy Global Weights $\{\Theta'_{r,i}\}_{i=1}^k := \Theta_r$
    **for** $i = 1$ **to** $k$ **do**

8       $\Theta''_{r,i} \leftarrow \Theta'_{r,i} - \eta_{\text{inner}} \nabla_{\Theta'_{r,i}} \mathcal{L}_{T_i}(\Theta'_{r,i})$

10     Compute Meta Gradients
    $[\nabla_{\theta_r^0}\mathcal{L}_{\text{meta}}, \ldots, \nabla_{\theta_r^{L-1}}\mathcal{L}_{\text{meta}}]$
    **for** $l = 0$ **to** $L - 1$ **do**

13       $g^l(r) = \mathbf{1}\{l \leq \lfloor \frac{rL}{\mathcal{R}} \rfloor\}$

15       $\theta_{r+1}^l \leftarrow \theta_r^l - \eta_{\text{outer}} g^l(r) \nabla_{\theta_r^l}\mathcal{L}_{\text{meta}}$

---

$$\nabla_{\Theta_r}\mathcal{L}_{\text{meta}} = \nabla_{\Theta_r} \sum_{i=1}^k \mathcal{L}_{\mathcal{T}_i}(\Theta''_{r,i})$$

$$\Theta_{r+1} = \Theta_r - \eta_{\text{outer}} \nabla_{\Theta_r}\mathcal{L}_{\text{meta}}$$

$$(3)$$

**Gated Layer-wise Optimization (GLO).** Traditional meta-learning approaches (Finn et al., 2017; Rajeswaran et al., 2019) are not physics-informed and treat all tasks in $\Gamma_{\text{easy}}$ as resulting in similar training dynamics across all layers. Thus, the meta-update in Eq. 3 is a simple summation of all task gradients. However, based on previous work (Wang et al., 2020), we know that non-trivial dynamics between the physics-based loss and data-driven loss affects different layers in the neural network differently. To address these gaps that manifest uniquely in the physics-informed PDE modeling context, we augment the IE procedure with the GLO mechanism by introducing a *gated* update of layers during each meta-update, such that a gating mechanism $g(r)$, *unlocks* deeper layers, exposing their parameters to gradient-based optimization, gradually, over the training iterations from 1 to $\mathcal{R}$.

Let us consider that the global PINN is an 'L' layer neural network whose parameters at iteration $r$ are represented as $\Theta_r = \{\theta_r^0, \ldots, \theta_r^{L-1}\}$ with $\theta_r^i$ representing the weights corresponding to layer $i$ at iteration $r$. Standard IE via. meta-learning (Finn et al., 2017) treats all layers identically during the gradient descent meta-update. However, via. GLO, we are able to prioritize learning shallow layers prior to deeper layers. Specifically, at each iteration $r \in \{1, 2, \ldots, \mathcal{R}\}$, the (binary) gating value for layer $l$ is calculated via. a linear gating mechanism as highlighted in Eq. 4.

$$g^l(r) = \begin{cases} 1, & l \leq \lfloor \frac{rL}{\mathcal{R}} \rfloor, \\ 0, & \text{otherwise,} \end{cases} \quad l = 0, \ldots, L-1. \tag{4}$$

The gating schedule begins with the first layer unmasked, followed by linear unmasking of deeper layers as training progresses, ensuring localized (and thereby more manageable) effects of non-trivial loss dynamics. Although the proposed $g(r) \in \{0,1\}^L$, mechanism results in a linear gating mechanism, GLO can easily admit any general gating mechanism.

$$\theta_{r+1}^l = \theta_r^l - \eta_{\text{outer}} \, g^l(r) \, \nabla_{\theta_r^l} \mathcal{L}_{\text{meta}}, \quad l = 0, \ldots, L-1, \tag{5}$$

If $g(r) = [g^0(r), \ldots, g^{L-1}(r)]$ and each meta-update gradient vector is represented as $\nabla_{\Theta_r} \mathcal{L}_{\text{meta}} = [\nabla_{\theta_r^0} \mathcal{L}_{\text{meta}}, \ldots, \nabla_{\theta_r^{L-1}} \mathcal{L}_{\text{meta}}]$, then the meta-update at iteration $r+1$ for layer $l$ is detailed in Eq. 5. Here, each $g^l(r)$ controls the meta-update strength of layer $l$. Alg. 1 showcases the entire procedure of our proposed LeIn-PINN framework for learning intelligent weight initialization. The LeIn weights $\Theta_{\mathcal{R}}$ obtained from the IE + GLO based learned initialization process can be employed as the initial weights for PINN training on $\gamma_{\text{hard}}$ (i.e., the *hard* PDE context).

## 4 Dataset description and Experimental Setup

We evaluate our proposed LeIn-PINN model on the three diverse PDE domains, described below.

**1D Convection.** The 1D convection PDE (Eq. 6) models transport of a scalar field $u(z,t)$ with constant velocity. In Eq. 6, $z \in [0, 2\pi]$ denotes the spatial coordinate and $t \in [0,1]$ denotes the temporal coordinate. $\beta$ is the advection coefficient, $u_t$ and $u_z$ denote partial derivatives in time and space. $u(z,0) = sin(z)$ defines the initial condition and periodic boundaries impose $u(0,t) = u(2\pi, t)$.

$$\mathcal{N}[u](z,t) := u_t(z,t) + \beta \, u_z(z,t) = 0, \tag{6}$$

This problem represents convection-dominated motion, where sharp discontinuities in the domain make PINN training challenging. In line with prior work Krishnapriyan et al. (2021), we consider $\Gamma_{\text{easy}}$ as $\beta \in \{5, 10, 15, 20, 25\}$ and hard regime as $\beta \in \{30, 40, 50, 60, 70, 80\}$.

**2D Helmholtz.** The steady-state Helmholtz PDE (Eq. 7) describes oscillatory wave fields on a 2D domain. Let $z = (x_1, x_2)$ and let $\Delta = \partial_{x_1 x_1} + \partial_{x_2 x_2}$ denote the Laplace operator. Eq. 7 is defined on the domain $[-1, 1]^2$ with Dirichlet boundary condition $u = 0$ on $\partial\Omega$. Using the analytic solution $u(z) = \sin(a_1 \pi x_1) \sin(a_2 \pi x_2)$, the source term becomes $q(z) = [k^2 - (a_1 \pi)^2 - (a_2 \pi)^2] \, u(z)$.

$$\mathcal{N}[u](z) := \Delta u(z) + k^2 u(z) - q(z) = 0, \tag{7}$$

The coefficients $(a_1, a_2)$ control the oscillation frequency along x and y direction respectively, and larger values induce more frequent oscillations posing greater challenge for PINNs. $\Gamma_{\text{easy}}$ is considered $\{(1,1), (1,2), (1,3), (1,4), (1,5)\}$ based on the case considered in Wang et al.

(2020) and hard regime is selected to have significantly more challenging propagation dynamics and both isotropic and anisotropic propagation dynamics are tested in the hard regimes $\{(4,4), (4,5), (5,5), (4,6), (5,6), (6,6)\}$. Our work has extended the investigation regime significantly, from previous PINN based investigations of 2D Helmholtz.

**2D Incompressible Navier–Stokes.** We finally consider modeling the challenging 2D viscous flow using the non-dimensional 2D incompressible Navier–Stokes equations, represented in Eq. 8 in compact momentum-continuity form. Here $\mathbf{u}(x,y,t)$ is the velocity field, $p(x,y,t)$ is the scalar pressure field, $\nu$ is the kinematic viscosity (parameterized through the Reynolds number). Domain $(x,y,t) \in \Omega \times [0,T]$, where $\Omega$ contains a unit-diameter cylinder.

$$\frac{\partial \mathbf{u}}{\partial t} + (\mathbf{u} \cdot \nabla)\mathbf{u} + \nabla p - \nu \nabla^2 \mathbf{u} = 0, \qquad \nabla \cdot \mathbf{u} = 0. \tag{8}$$

This system exhibits vortex shedding and transitional flow behavior, and the coupling between momentum and incompressibility creates multiple interacting residual terms, making it one of the most challenging PDE families for PINNs. The complexity of the dynamics increases with increase in Reynolds number $Re$. We consider $\Gamma_{\text{easy}}$ to be $Re \in \{100, 200, 300, 400, 500\}$ where relatively more steady, laminar dynamics are exhibited and in line with Lee et al. (2025), we consider $Re \in \{600, 800, 1000\}$ to be hard regime where less steady dynamics are exhibited.

## 5 RESULTS & DISCUSSION

In this section, we demonstrate with rigorous quantitative and qualitative experiments, the effects of catastrophic failures on PINN performance and how these failures can be alleviated via. Learned Initialization (LeIn). To validate our approach, we compare LeIn-PINN against a suite of state-of-the-art models, that have been designed specifically to overcome catastrophic failures of PINNs in challenging PDE domains; the full descriptions of these baselines can be found in Appendix D.2. Specifically, we undertake this investigation by answering three research questions (RQ):

**RQ1.** How does learned initialization alleviate catastrophic training dynamics in challenging PDE domains?

**RQ2.** How do LeIn-PINNs compare with other state-of-the-art (SoTA) approaches proposed to alleviate PINN catastrophic failures?

**RQ3.** What is the effect of invariance encoding and gated layer-wise optimization in LeIn-PINN? (Ablation Analysis)

We evaluate our approach on three representative PDE systems: the 1D convection equation, the 2D Helmholtz equation, and the 2D time-dependent incompressible Navier–Stokes equations. Full specifications of each setup, including PDE definitions, boundary and initial conditions, PDE Parameters, and training configurations, are provided in Section 4, while additional training information such as model hyperparameters and architectural details is included in the Appendix. D.

### 5.1 (RQ1) HOW DOES LEARNED INITIALIZATION ALLEVIATE CATASTROPHIC TRAINING DYNAMICS IN CHALLENGING PDE DOMAINS?

We answer this question in three stages. First, we demonstrate qualitatively the catastrophic failures PINNs undergo in challenging PDE contexts. Second, we investigate how the training losses of the PINNs evolve in challenging domains. Third, we characterize the effect of the catastrophic training dynamics by analyzing the loss landscape neighborhood of the converged models. In each case, we contrast the analysis with the behavior of LeIn-PINN in the same context.

**Qualitative Failure Modes.** In Fig. 1a, we demonstrate the solution predicted by a PINN (with random weight initialization) when trained on a challenging 1D convection domain with $\beta = 70$. It can be clearly seen that the PINN model (center plot in Fig. 1a) has failed to estimate a good solution of the domain. The perfect solution of the domain can be seen in the left plot (labeled 'Analytical Solution') of Fig. 1a. We conduct a similar qualitative investigation of a separate PDE domain 2D-Helmholtz comprising two PDE parameters $a_1, a_2$, with $(a_1 = 6, a_2 = 6)$ in Fig. 1b. Once again, we notice that the PINN model (center plot of Fig. 1b) is unable to provide a good estimate of the challenging PDE domain solution (i.e., Analytical Solution represented on the left plot in Fig. 1b). Turning our attention to the performance of LeIn-PINN in each of these two contexts (i.e., right plot

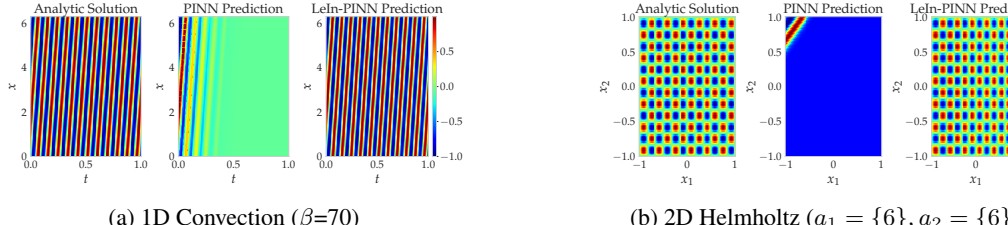

(a) 1D Convection ($\beta$=70)

(b) 2D Helmholtz ($a_1 = \{6\}, a_2 = \{6\}$)

Figure 1: Qualitative performance comparison of a randomly initialized PINN model and *intelligently initialized* PINNs i.e., LeIn-PINN on two different and challenging PDE domains. We notice that both in the 1D convection Fig. 1(a) and 2D Helmholtz Fig. 1(b), LeIn-PINN faithfully recreates the analytical solution while the randomly initialized PINN variant experiences catastrophic failure.

in each of Fig. 1a and Fig. 1b), we notice that the estimation of LeIn-PINN closely aligns with the analytical solution of the corresponding PDE domain. We now highlight that the only difference between the PINN and LeIn-PINN procedures is the weight initialization. Specifically, PINN weights have been initialized randomly (i.e., Xavier initialization) while the weights of LeIn-PINN have been initialized by our proposed *learned initialization* method. Once weights have been initialized, both models have been subjected to identical training procedures on the target PDE domain. Thus, we infer that the learned initialization strategy in LeIn-PINN alleviates catastrophic failures even in challenging PDE domains.

**Extension to Navier-Stokes.** As a qualitative stress test in a multi-physics regime, we extend our analysis to the 2D incompressible, time-dependent Navier-Stokes cylinder wake at high Reynolds number. We supervise only velocity components $(u, v)$, while pressure is reconstructed in an unsupervised manner. In the inverse setting, the advection and viscosity coefficients $(\lambda_{\text{adv}}, \lambda_{\text{vis}})$ are treated as learnable parameters. Consistent with our findings on 1D-Convection and 2D-Helmholtz, LeIn-PINN yields substantially lower errors and qualitatively faithful pressure fields compared to randomly initialized PINN.

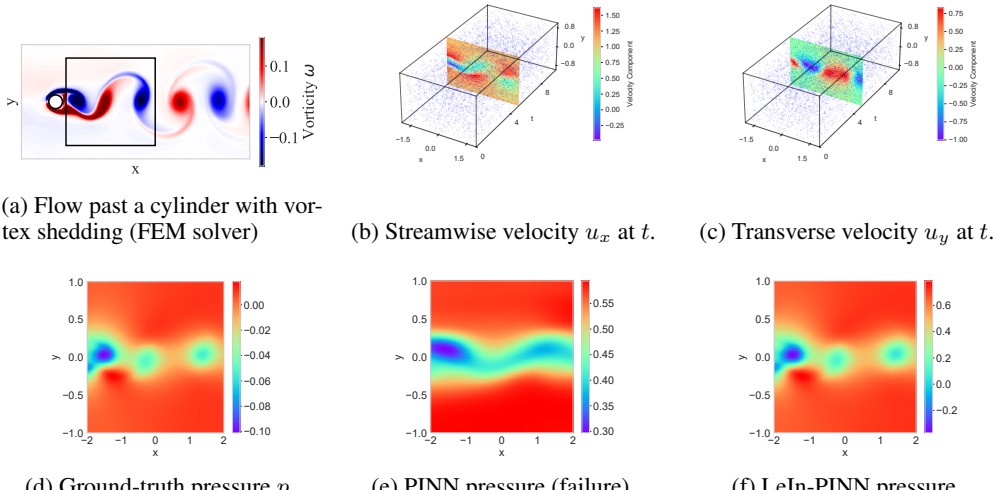

(a) Flow past a cylinder with vortex shedding (FEM solver)

(b) Streamwise velocity $u_x$ at $t$.

(c) Transverse velocity $u_y$ at $t$.

(d) Ground-truth pressure $p$.

(e) PINN pressure (failure).

(f) LeIn-PINN pressure.

Figure 2: Navier-Stokes cylinder wake at $\text{Re} = 1000$. (a) Simulation setup with vortex shedding (FEM solver). The highlighted box indicates the region used for training and evaluation. (b–c) Velocity components $u_x, u_y$ at a representative time slice. Also showcased are collocation points at which velocity training data is sampled. (d) Ground-truth pressure. (e) Standard PINN prediction (failure). (f) LeIn-PINN prediction, showing faithful reconstruction.

In Fig. 2, we clearly see that vanilla PINNs fail to reproduce the pressure field Fig. 2(e), while LeIn-PINN reconstructs the wake structures faithfully Fig. 2(f). This highlights the advantage of

learned initialization in complex fluid dynamics.

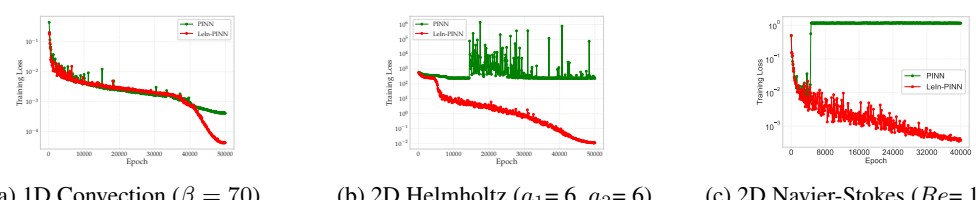

    (a) 1D Convection ($\beta = 70$)    (b) 2D Helmholtz ($a_1= 6$, $a_2= 6$)    (c) 2D Navier-Stokes ($Re= 1000$)

Figure 3: Training-loss curves highlighting training dynamics of randomly initialized PINNs and LeIn-PINN, across three separate PDE domains. We notice in all three cases that LeIn-PINN converges to a solution with (at least) an order of magnitude lower loss, compared to randomly initialized PINN.

**Training Dynamics Analysis.** We complement the qualitative analysis in Fig. 1 with an investigation (Fig.3) of the training dynamics in the same PDE contexts. Specifically, we highlight in Fig.3, the evolution of the total training loss (Eq. 15) of PINNs and LeIn-PINN on the same 1D convection, 2D Helmholtz and 2D Navier-Stokes domains. We note that the training loss plots in Fig.3 has been averaged over five independent training runs to capture the *expected training dynamics*. Loss plots of individual training runs are in appendix E.2.

For the 1D convection problem ($\beta = 70$) the result depicted in Fig. 3a, shows that random initialization of the PINN leads to an early plateau in the loss curve, whereas our learned initialization method (LeIn-PINN) achieves a steady, monotonic decrease, signaling more effective exploration of the solution manifold and ultimately, convergence to a loss value an order of magnitude lower than the corresponding randomly initialized PINN solution. A similar pattern emerges on the 2D Helmholtz and 2D Navier-Stokes problem ($a_1 = 6$, $a_2 = 6$) as shown in Fig. 3b. We see that the randomly initialized PINN exhibits premature convergence to a sub-optimal solution while, once again LeIn-PINN achieves a loss *several orders of magnitude lower*. This demonstrates the ability of our learned initialization based solution (LeIn-PINN) to achieve superior convergence dynamics by overcoming loss surface barriers Fort et al. (2020) and hence avoiding catastrophic failures.

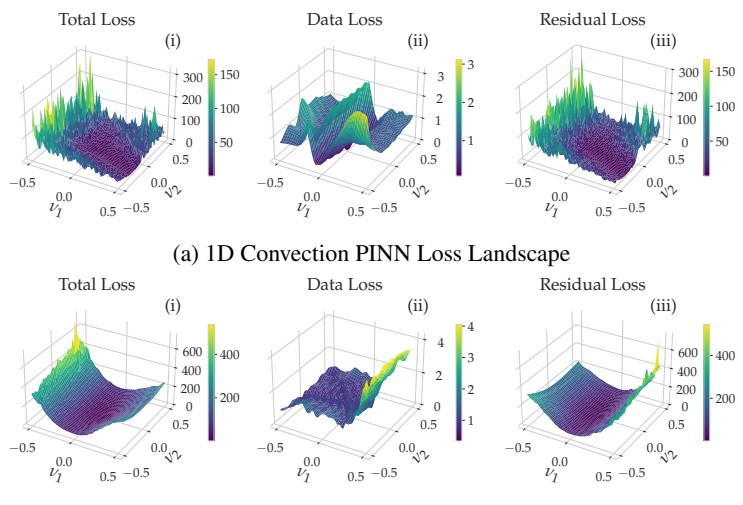

(a) 1D Convection PINN Loss Landscape

(b) 1D Convection LeIn-PINN Loss Landscape

Figure 4: Loss Landscape visualization generated by perturbing converged model weights along its top-2 eigenvectors ($\nu_1$, $\nu_2$). (a) Randomly Initialized PINN 1D Convection, with (i), (ii), (iii) representing loss landscape w.r.t total loss, data loss and residual loss respectively. (b) LeIn-PINN 1D Convection with (i), (ii), (iii) once again representing loss landscape w.r.t total loss, data loss and residual loss respectively.

**Hessian Eigenvector Landscape Analysis.** Although the loss curves in Fig. 3 show a clear contrast between random and learned initialization, they do not fully reveal how training dynamics affect parameter optimization. To clarify this, we examine the loss landscape neighborhood of each *converged* model using a Hessian-informed perturbation analysis (Krishnapriyan et al., 2021; Böttcher & Wheeler, 2024). Fig. 4 presents results for the 1D convection problem (2D Helmholtz and 2D Navier-Stokes in Appendix C). Under random initialization (Fig. 4a), the converged PINN exhibits a highly rugged surface with deep valleys and sharp ridges, driven mainly by the residual term (Fig. 4a.iii). The presence of such ravines is known to hinder convergence to strong local minima Wu et al. (2017); Hochreiter & Schmidhuber (1997). In contrast, the data loss landscape is smoother but still challenging to optimize. With LeIn-PINN, the converged landscape is markedly smoother and lacks the sharp peaks of Fig. 4a. Fig. 4b.i shows the total loss surface, while Figs. 4b.ii–b.iii show data and residual components, all with reduced contrast and smoother geometry compared to their PINN counterparts. Viewed together with Fig. 3a, these results illustrate that random initialization produces residual-dominated peaks that trap the optimizer in narrow basins, whereas LeIn-PINN yields a smoother landscape that supports improved convergence to better local minima.

### 5.2 (RQ2) How do LeIn-PINNs compare with other state-of-the-art (SoTA) approaches proposed to alleviate PINN catastrophic failures?

Recent efforts have begun to investigate catastrophic failures in PINNs, and in this section we compare the performance of our learned initialization based LeIn-PINN model relative to three recent state-of-the-art approaches (Krishnapriyan et al., 2021; Wang et al., 2020; Daw et al., 2023; Qin et al., 2022) (detailed in Sec. D.2), explicitly designed with the intention of alleviating such failures. We provide rigorous quantitative comparisons across three benchmark PDE systems: the 1D Convection equation, the 2D Helmholtz equation, and the 2D incompressible, time-dependent Navier–Stokes system (Raissi et al., 2017b). In all cases, we evaluate models under challenging *extrapolation* settings, where PDE parameters or regimes lie outside those directly observed during initialization. For Navier–Stokes, our formulation jointly addresses both the *forward problem* (reconstructing velocity and pressure fields) and the *inverse problem* (learning unknown PDE coefficients such as advection and viscosity) within a single PINN framework.

Table 1: Mean Absolute Error (MAE) on the 1D convection system for extrapolation tasks ($\beta \in \{30, 40, 50, 60, 70, 80\}$).

| Method | 30 | 40 | 50 | 60 | 70 | 80 | **Avg** |
|---|---|---|---|---|---|---|---|
| PINN (fixed) | 0.8234 | 0.8461 | 0.7926 | 0.7972 | 0.6587 | 0.8732 | 0.7985 |
| PINN (dynamic) Wang et al. (2021a) | 0.0071 | 0.0126 | 0.0226 | 0.1607 | 0.3670 | 0.4634 | 0.1722 |
| Curr.-Reg. Krishnapriyan et al. (2021) | 0.1403 | 0.2719 | 0.3604 | 0.3929 | 0.4566 | 0.4695 | 0.3486 |
| R3 Daw et al. (2023) | **0.0067** | 0.0678 | 0.3224 | 0.4319 | 0.4726 | 0.4988 | 0.3000 |
| Meta-PDEQin et al. (2022) | 0.04336 | 0.1588 | 0.2112 | 0.3757 | 0.35702 | 0.5167 | 0.2771 |
| LeIn-PINN | 0.0092 | **0.0122** | **0.0150** | **0.0172** | **0.0272** | **0.2490** | **0.0550** |

**1D Convection System** LeIn-PINN outperforms all baselines and vanilla PINNs (Raissi et al., 2017b). Table 1 shows that vanilla PINNs often collapse due to unstable training, while Curriculum-Reg and Adaptive Sampling offer limited improvements but degrade under extrapolation. Across all comparative baselines, LeIn-PINN achieves a reduction in MAE ranging from **68% to 93%**, showing that initialization alone can surpass engineered training strategies. To further verify that the increase in performance is inherent to LeIn weights and not just number of gradient descent steps, we conduct an analysis where all models are trained with same number of gradient descent updates as LeIn-PINN (Invariance Encoding + Task Adaptation). We find that LeIn-PINNs still achieve errors an order of magnitude lower across all extrapolation contexts.This analysis is included in Appendix E.8.

**2D Helmholtz System** On Helmholtz, baselines fail to capture oscillatory solutions. As shown in Table 2, LeIn-PINN provides consistently large gains, improving MAE by **95% to 99%** across all comparative baselines, confirming robustness on high-frequency, multi-parameter PDEs.

**2D Incompressible Navier-Stokes** Table 3 and 4] reports forward and inverse settings. For forward pressure reconstruction, LeIn-PINN outperforms all comparative baselines by **94% to 98%**, highlighting its ability to stabilize training in turbulent flow regimes. In the inverse setting, LeIn-PINN yields improvements ranging from **20% to 97%** across all comparative baselines, with the largest gains observed in advection and pressure estimation.

Table 2: Mean Absolute Error (MAE) on the 2D Helmholtz PDE across varying domain parameters $a_{xy}$. Baseline methods are shown as rows for clearer comparison.

| Method | $a_{44}$ | $a_{45}$ | $a_{55}$ | $a_{46}$ | $a_{56}$ | $a_{66}$ | **Avg** |
|---|---|---|---|---|---|---|---|
| PINN (fixed) | 0.3785 | 0.4956 | 0.5838 | 1.0264 | 0.9753 | 1.0783 | 0.7563 |
| PINN (dynamic) Wang et al. (2021a) | 0.0047 | 0.2154 | 0.1962 | 0.0850 | 0.0879 | 0.1687 | 0.1263 |
| Curr-Reg. Krishnapriyan et al. (2021) | 0.4131 | 0.6172 | 0.4106 | 0.4050 | 0.4390 | 0.4346 | 0.4533 |
| R3 Daw et al. (2023) | 0.3864 | 1.5196 | 0.4173 | 0.4272 | 0.4139 | 0.4204 | 0.5975 |
| Meta-PDEQin et al. (2022) | 0.2473 | 0.4736 | 0.5982 | 0.4954 | 0.4077 | 0.4329 | 0.4425 |
| LeIn-PINN | **0.0013** | **0.0029** | **0.0058** | **0.0050** | **0.0081** | **0.0150** | **0.0064** |

Table 3: Forward problem: Pressure ($p$) prediction errors across Reynolds numbers. Lowest values in each column are in bold.

| Method | 600 | 800 | 1000 | Avg. |
|---|---|---|---|---|
| PINN (fixed) | 0.2456 | 0.0137 | 0.0426 | 0.1006 |
| PINN (dynamic) Wang et al. (2021a) | 0.2516 | 0.0073 | 0.0059 | 0.0883 |
| Curr-Reg. Krishnapriyan et al. (2021) | 0.2133 | 0.2150 | 0.2147 | 0.2143 |
| R3 Daw et al. (2023) | 0.2399 | 0.2507 | 0.1349 | 0.2086 |
| Meta-PDEQin et al. (2022) | 0.3172 | 0.2997 | 0.2812 | 0.2994 |
| LeIn-PINN | **0.0057** | **0.0058** | **0.0057** | **0.0057** |

Table 4: Inverse problem: Estimation errors for $\lambda_{\text{advection}}$ and $\lambda_{\text{viscosity}}$ across Reynolds numbers. Lowest values in each column are in bold.

| Method | 600 | | 800 | | 1000 | | Avg. | |
|---|---|---|---|---|---|---|---|---|
| | $\lambda_{\text{adv}}$ | $\lambda_{\text{visc}}$ | $\lambda_{\text{adv}}$ | $\lambda_{\text{visc}}$ | $\lambda_{\text{adv}}$ | $\lambda_{\text{visc}}$ | $\lambda_{\text{adv}}$ | $\lambda_{\text{visc}}$ |
| PINN (fixed) | 0.2067 | 0.0729 | 0.0206 | 0.0074 | 0.0618 | 0.0258 | 0.0964 | 0.0354 |
| PINN (dynamic) Wang et al. (2021a) | 0.2164 | 0.0112 | 0.0113 | 0.0077 | 0.0083 | 0.0069 | 0.0787 | 0.0086 |
| Curr-Reg. Krishnapriyan et al. (2021) | 0.3970 | 0.0845 | 0.3971 | 0.0843 | 0.3988 | 0.0841 | 0.3976 | 0.0843 |
| R3 Daw et al. (2023) | 0.4731 | 0.0497 | 0.3852 | 0.0860 | 0.2741 | 0.0419 | 0.3774 | 0.0592 |
| Meta-PDEQin et al. (2022) | 0.4320 | 0.1312 | 0.6910 | 0.2979 | 0.6910 | 0.2370 | 0.6046 | 0.2220 |
| LeIn-PINN | **0.0083** | **0.0068** | **0.0085** | **0.0071** | **0.0081** | **0.0068** | **0.0083** | **0.0069** |

Overall, LeIn-PINN averages a **89%** improvement over SoTA, demonstrating benefits for predictive accuracy and PDE coefficient recovery in fluid dynamics.

### 5.3 (RQ3) What is the effect of invariance encoding and gated layer-wise optimization in LeIn-PINN? (Ablation Analysis)

The learned initialization aspect of LeIn-PINNs has two primary components, (i) *Invariance Encoding* (IE) (ii) Gated Layer-wise Optimization (GLO). To test the effect of each of these components, we investigate the performance of *ablation variants*, 'LeIn-PINN w/o (IE,GLO)', which is a variant of LeIn-PINN without IE or GLO; and 'LeIn-PINN w/o GLO', variant of LeIn-PINN without GLO which essentially boils down to using a generic MAML Finn et al. (2017) model. LeIn-PINN w/o (IE,GLO), which is nothing but a randomly initialized PINN trained on the PDE domain of interest. The ablation analysis is carried out by inspecting (i) performance volatility, which measures median, inter-quartile range and standard deviation of quantitative performance across multiple runs of each model across multiple tasks of varied complexity; (ii) spectral bias, which measures the extent to which each ablation variant and LeIn-PINN are biased towards capturing low-frequency components of the domain while ignoring high-frequency components.

**Analyzing Performance Volatility.** In Fig. 5, we examine the impact of different initialization strategies on model performance by training to convergence and plotting the distribution of mean absolute errors for each method. Fig. 5a and Fig. 5b, showcase that the variant LeIn-PINN w/o (IE, GLO) exhibits high volatility (i.e., large interquartile range) as well as high median error with increasing task complexity while the variants with learned initialization ( LeIn-PINN w/o GLO and LeIn-PINN) exhibit lower degradation in median performance as well as lower inter-quartile

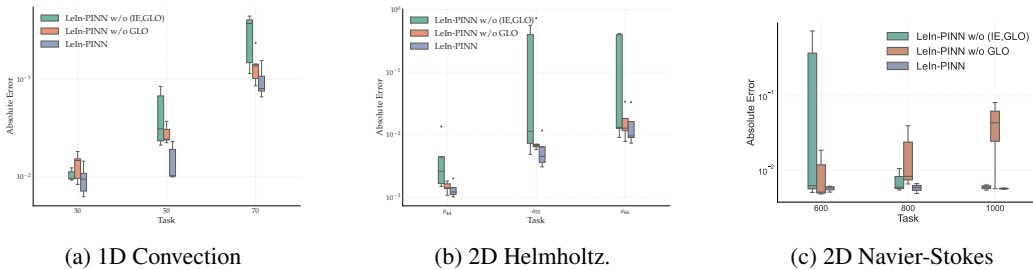

(a) 1D Convection      (b) 2D Helmholtz.      (c) 2D Navier-Stokes

Figure 5: Model-comparison boxplots for PINN convergence on **(a)** 1D Convection, **(b)** 2D Helmholtz and **(c)** 2D Navier-Stokes. Errors reported in log scale on uniformly sampled sub-set of tasks for ease of reading; results on unreported tasks are similar.

ranges. Together, these box plots illustrate two stages of improvement: first, IE reduces performance volatility compared to random initialization i.e., LeIn-PINN w/o (IE,GLO); furtuer, GLO further lowers both median error as well as performance volatility, ensuring consistently reliable PINN training. To examine the robustness of LeIn-PINN, we analyzed its sensitivity to both the number and distribution range of sampled tasks, confirming stable performance across diverse task configurations. Our initialization strategy introduces only a one-time meta-training overhead, after which LeIn-PINN scales comparably to standard PINNs in both training and inference cost. The detailed robustness and compute analyses are provided in Appendix E.7 and Appendix E.9, respectively.

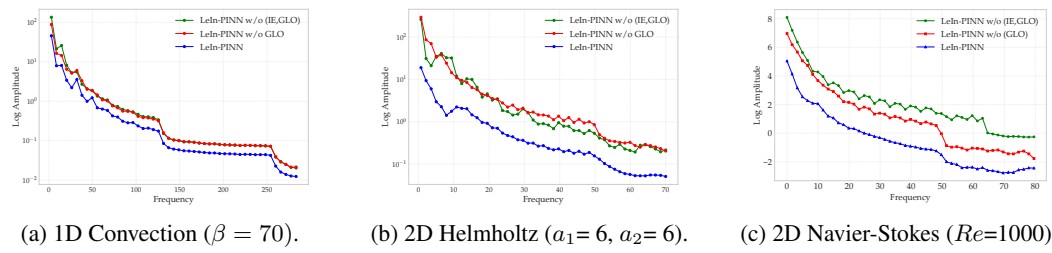

(a) 1D Convection ($\beta = 70$).    (b) 2D Helmholtz ($a_1 = 6$, $a_2 = 6$).    (c) 2D Navier-Stokes ($Re$=1000).

Figure 6: Comparison of spectral distribution of LeIn-PINN and ablation variants. We notice that LeIn-PINN alleviates spectral bias owing to GLO and IE across 1D convection, 2D Helmholtz and 2D Navier-Stokes.

**Spectral Bias Analysis.** Neural networks often exhibit spectral bias, learning low-frequency components faster than high-frequency ones. To quantify this, we compute the Fourier transform of the absolute error over the spatial domain and plot amplitude spectra for each initialization strategy (Fig. 6). Both LeIn-PINN variants without GLO and without (IE,GLO) show similar spectral density, while full LeIn-PINN (with IE and GLO) alleviates spectral bias significantly. Since Fig. 6 visualizes spectral density of absolute error fields, lower values are better. We observe that LeIn-PINN achieves consistently lower density across low and high frequencies in 1D convection (a), 2D Helmholtz (b) and 2D Navier-Stokes(c). Standard Xavier and LeIn-PINN (w/o GLO) exhibit pronounced high-frequency error, whereas LeIn-PINN yields substantially lower error across the entire band, capturing fine-scale features more effectively. Beyond this single case, Appendix E.3 reports mean energy density per task across all PDEs, again showing that LeIn-PINN reduces spectral bias relative to ablation variants, highlighting the effect of invariance encoding and gated layer-wise optimization.

# 6 CONCLUSION

In this work, we introduced LeIn-PINN, a novel variant of physics-informed neural networks with learned initialization. Through rigorous qualitative and quantitative experiments, we demonstrated that LeIn-PINN alleviates catastrophic failures commonly observed in PINNs when applied to challenging PDE domains. Our comparisons with state-of-the-art methods highlight the superiority of LeIn-PINN, achieving an average performance improvement of **89%** across multiple PDE systems and baselines. In addition, our ablation analysis shows that the invariance encoding and gated layer-wise optimization procedures in LeIn-PINN reduce performance volatility and mitigate spectral bias in PINN models across simple and complex PDE tasks.

## REPRODUCIBILITY STATEMENT

We have made efforts to ensure that our results are reproducible. The main text provides a clear description of our model. Additional implementation details, including architecture specifications, hyperparameters, and training schedules, are provided in the Appendix D.3. As the full datasets are too large to share directly, we instead provide dataset generation code. We also include experiment scripts, along with the PINN implementation, to allow reproduction of all reported results and figures using our methodology.

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

## A   LIMITATIONS

Despite the strong empirical gains demonstrated across our benchmark PDEs, our study has the following limitations:

- **Analysis on broader system dynamics.** While we evaluated both time-dependent and time-independent PDEs, we have not yet extended our approach to truly chaotic regimes (e.g. the chaotic phase of the Kuramoto–Sivashinsky equation) or PDEs with higher-order terms. Such systems pose unique challenges for optimization and stability, and remain important directions for future work.

- **Lack of formal theoretical guarantees.** Our conclusions are based on extensive empirical evidence, including Hessian- and gradient-based analyses across layers. However, we do not yet provide a formal mathematical proof for the effectiveness of our learned initialization. Developing such a theoretical foundation is part of our ongoing research.

- **Fixed training task schedule.** The meta-training process uses a predetermined set of tasks without exploring adaptive or rotational sampling strategies. How dynamically adjusting task difficulty or sampling schedules might affect convergence and the quality of learned initialization remains an open question.

## B   DETAILED FORMULATION OF PINN

In this section, we first present the general formulation of Physics-Informed Neural Networks (PINNs), and then demonstrate its instantiation on two canonical PDEs: the 1D Convection Equation and the 2D Helmholtz Equation.

### B.1   PHYSICS INFORMED NEURAL NETWORK

A general form of a PDE can be expressed as:

$$\mathcal{F}(u(\mathbf{z}); \gamma) = f(\mathbf{z}), \quad \mathbf{z} \in \Omega, \tag{9}$$

$$\mathcal{B}(u(\mathbf{z})) = g(\mathbf{z}), \quad \mathbf{z} \in \partial\Omega. \tag{10}$$

Here:

- $\Omega \subset \mathbb{R}^d$ represents the spatial domain, while $\partial\Omega$ denotes its boundary.
- $\mathbf{z} = [x_1, x_2, \ldots, x_d, t]^\top \in \mathbb{R}^{d+1}$ encapsulates space-time coordinates.
- $u$ is the unknown solution we aim to approximate.
- $\gamma$ denotes physical parameters of the system.
- $f(\mathbf{z})$ encodes system-specific data or forcing terms.
- $\mathcal{F}$ is a (potentially nonlinear) differential operator characterizing the physical laws of the system.
- $\mathcal{B}$ is a boundary operator that enforces initial and boundary conditions, which can be of Dirichlet, Neumann, or periodic types.

The PINN framework leverages automatic differentiation to embed the residuals of $\mathcal{F}$ and $\mathcal{B}$ into a composite loss function. This loss function penalizes deviations from the governing equations and boundary conditions, ensuring that the solution adheres to the physical laws while simultaneously fitting observed data.

Neural networks are powerful tools due to their universal approximation property, which allows them to approximate any continuous function given sufficient capacity. In the context of PINNs, we aim to approximate the solution $u(\mathbf{z})$ of the governing equations using a neural network $\hat{u}_\theta(\mathbf{z})$, where $\theta$ denotes the learnable parameters of the neural network. The approximation can be written as:

$$\hat{u}_\theta(\mathbf{z}) \approx u(\mathbf{z}), \tag{11}$$

where $\mathbf{z} \in \mathbb{R}^{d+1}$ represents the space-time coordinates.

To achieve this, the PINN framework constructs a composite loss function consisting of:

1. PDE Residual Loss: This enforces the governing physical laws as specified by the PDE.

2. Boundary/Data Loss: This ensures compliance with boundary or initial conditions, as well as data consistency when available.

**PDE Residual Loss**    Using automatic differentiation tools, the neural network computes derivatives efficiently with respect to its inputs, enabling evaluation of the PDE residual. The residual is given by:

$$\mathcal{R}(\mathbf{z}) = \mathcal{F}(\hat{u}_\theta(\mathbf{z}); \gamma) - f(\mathbf{z}), \tag{12}$$

where $\mathcal{F}$ represents the differential operator, $f(\mathbf{z})$ is the system-specific forcing term, and $\gamma$ are the physical parameters. The residual loss over the domain $\Omega$ is formulated as:

$$\mathcal{L}_r = \frac{1}{N_\Omega} \sum_{i=1}^{N_\Omega} |\mathcal{R}(\mathbf{z}_i)|^2, \tag{13}$$

where $N_\Omega$ represents the number of collocation points in the domain, and $\mathbf{z}_i \in \Omega$ are the sampled points.

**Boundary/Data Loss**    For enforcing boundary or initial conditions, we define a loss term that penalizes deviations from the specified boundary values $g(\mathbf{z})$. For Dirichlet conditions, this can be written as:

$$\mathcal{L}_b = \frac{1}{N_{\partial\Omega}} \sum_{j=1}^{N_{\partial\Omega}} |\mathcal{B}(\hat{u}_\theta(\mathbf{z}_j)) - g(\mathbf{z}_j)|^2, \tag{14}$$

where $N_{\partial\Omega}$ represents the number of points on the boundary $\partial\Omega$, and $\mathbf{z}_j \in \partial\Omega$ are the boundary points.

**Total Loss**    The total loss for training the PINN combines the residual loss and the boundary/data loss, weighted by respective coefficients $\lambda_r$ and $\lambda_d$:

$$\mathcal{L}_{total} = \lambda_r \mathcal{L}_r + \lambda_d \mathcal{L}_b. \tag{15}$$

We calculate the total loss by adding the residual and data losses. Here, $\lambda_r$ and $\lambda_d$ are hyperparameters that control the relative contribution of the residual and data losses during training. The residual loss acts as a regularization term, ensuring that the learned solution adheres to physically consistent laws.

To train the neural network, we solve the optimization problem:

$$\theta^* = \arg\min_\theta \mathcal{L}_{total}. \tag{16}$$

We aim to formulate a Physics-Informed Neural Network (PINN) by studying two fundamental classification of systems in physics that are crucial for understanding and modeling the physical world. These systems are categorized based on whether their properties depend on time:

Time-Independent Systems These systems are used to study the steady-state behavior of a system, where properties such as energy, momentum, or other conserved quantities remain constant over time. Time-independent systems help us analyze static or equilibrium conditions, providing insights into the inherent properties of the system without considering temporal evolution.

Time-Dependent Systems These systems are used to study how the state of a system evolves over time. By understanding the temporal dynamics, we can gain insights into transient phenomena, system responses, and changes in state variables due to external forces or intrinsic behaviors.

To explore these concepts, we explore the following two systems.

### B.1.1    1D CONVECTION EQUATION

This system represents a classic example of a time-dependent system. We consider a one-dimensional convection system to demonstrate the application of Physics-Informed Neural Networks (PINNs). The system models the transport of heat or mass in a medium, where the primary variable $u(x, t)$ evolves over space and time. The spatial domain is $x \in [0, 2\pi]$, and the time domain is $t \in [0, 1]$. This

setup allows for studying convection behavior under periodic boundary conditions, often encountered in cyclic systems. This formulation is adapted from Krishnapriyan et al. (2021)

The governing partial differential equation (PDE) is given by:

$$\frac{\partial u}{\partial t} + \beta \frac{\partial u}{\partial x} = 0, \quad x \in \Omega, \, t \in [0, T], \tag{17}$$

with the initial condition:

$$u(x, 0) = h(x), \quad x \in \Omega. \tag{18}$$

Here:

- $\beta$ is the convection coefficient (physical parameter of system $\gamma$ in 9),
- $h(x)$ specifies the initial condition for $u(x, t)$,
- $\Omega$ denotes the spatial domain.

To evaluate the performance of PINNs, we generate ground truth data using an analytical solution. For constant $\beta$ and periodic boundary conditions, the analytical solution is:

$$u_{analytical}(x, t) = \mathcal{F}^{-1}\left[\mathcal{F}(h(x))e^{-i\beta kt}\right], \tag{19}$$

where:

- $\mathcal{F}$ and $\mathcal{F}^{-1}$ are the Fourier transform and its inverse, respectively,
- $i = \sqrt{-1}$ is the imaginary unit,
- $k$ represents the frequency in the Fourier domain.

For this problem, we assume:

$$h(x) = \sin(x), \quad u(0, t) = u(2\pi, t), \tag{20}$$

implying periodic boundary conditions.

**Residual Loss**   The residual loss for this PDE is given by:

$$\mathcal{L}_r = \frac{1}{N_f} \sum_{i=1}^{N_f} \left(\frac{\partial \hat{u}}{\partial t} + \beta \frac{\partial \hat{u}}{\partial x}\right)^2, \tag{21}$$

where:

- $N_f$: Number of collocation points sampled from the interior of the domain $\Omega \times [0, T]$
- $\hat{u}$ is the predicted solution,
- $\frac{\partial \hat{u}}{\partial t}$: Temporal derivative of the predicted solution,
- $\frac{\partial \hat{u}}{\partial x}$: Spatial derivative of the predicted solution,
- $\beta$: Convection coefficient which is the parameter describing the physical system

**Data Loss**   The data loss consists of two components: the loss enforcing the initial condition and the loss enforcing the boundary conditions. It is defined as:

$$\mathcal{L}_d = \mathcal{L}_{ic} + \mathcal{L}_{bc}, \tag{22}$$

where:

$$\mathcal{L}_{ic} = \frac{1}{N_u} \sum_{i=1}^{N_u} \left(\hat{u}(x_i, 0) - g(x_i, 0)\right)^2, \tag{23}$$

$$\mathcal{L}_{bc} = \frac{1}{N_b} \sum_{j=1}^{N_b} \left(\hat{u}(0, t_j) - \hat{u}(2\pi, t_j)\right)^2. \tag{24}$$

Here:

- $\mathcal{L}_{ic}$: Initial condition loss, ensuring the predicted solution $\hat{u}(x,0)$ matches the data $g(x,0) = h(x)$,

- $\mathcal{L}_{bc}$: Boundary condition loss, ensuring periodic boundary conditions $\hat{u}(0,t) = \hat{u}(2\pi,t)$,

- $N_u$: Number of points sampled from the initial condition,

- $N_b$: Number of points sampled from the boundary condition,

- $g(x_i,0) = h(x_i)$: Known solution at initial condition points,

- $\hat{u}(0,t_j)$ and $\hat{u}(2\pi,t_j)$: Predicted solutions at boundary points.

**Total Loss**    The total loss combines the above components:

$$\mathcal{L}(\theta) = \lambda_r \mathcal{L}_r + \lambda_d \mathcal{L}_d, \tag{25}$$

where:

- $\lambda_r, \lambda_d$ are weighting coefficients balancing the contributions of the respective terms.

### B.1.2    2D HELMHOLTZ EQUATION

We study the vibration states of a two-dimensional membrane using the 2D Helmholtz equation. This equation is widely used to describe stationary wave fields, such as vibrations in membranes, sound waves, and electromagnetic waves. Specifically, we consider a square membrane with spatial coordinates $x_1, x_2 \in [-1, 1]$ and analyze its steady-state behavior under fixed boundary conditions. The Helmholtz equation helps identify resonance patterns and steady-state vibration modes in such systems. The formulation is adapted from Wang et al. (2020).

The governing PDE for the 2D Helmholtz equation is:

$$\frac{\partial^2 u}{\partial x_1^2} + \frac{\partial^2 u}{\partial x_2^2} + k^2 u = q(x_1, x_2), \quad (x_1, x_2) \in \Omega, \tag{26}$$

where $\Omega = [-1, 1] \times [-1, 1]$ is the spatial domain.

The boundary of the domain, $\partial\Omega$, represents the fixed edges of the membrane. Consequently, the displacement $u(x_1, x_2)$ at the boundaries is zero, giving rise to the Dirichlet boundary condition:

$$u(x_1, x_2) = 0, \quad (x_1, x_2) \in \partial\Omega. \tag{27}$$

We specifically choose $k^2 \neq a_1^2 + a_2^2$ to ensure that the selected $k$ does not correspond to an eigenmode of the domain, resulting in a non-zero source term $q(x_1, x_2)$. Thus, the PDE system is inherently inhomogeneous.

Here:

- $k$ is the wavenumber and is kept fixed at $k = 1$.

- $a_1, a_2 \in \mathbb{Z}^+$ are integer mode numbers representing the spatial oscillation patterns. These serve as the primary PDE parameters affecting solution complexity and difficulty.

- $q(x_1, x_2)$ is the resulting source term that characterizes external excitation or forcing within the membrane, driving the system response.

For this setup, we use an analytical solution of the form:

$$u(x_1, x_2) = \sin(a_1 \pi x_1) \sin(a_2 \pi x_2), \tag{28}$$

which satisfies the zero-displacement boundary condition. The corresponding source term is derived as:

$$q(x_1, x_2) = \left[ k^2 - (a_1\pi)^2 - (a_2\pi)^2 \right] \sin(a_1 \pi x_1) \sin(a_2 \pi x_2). \tag{29}$$

**Residual Loss**  The residual loss quantifies deviations from the Helmholtz equation at collocation points in $\Omega$:

$$\mathcal{L}_{\text{residual}} = \frac{1}{N_f} \sum_{i=1}^{N_f} \left( \frac{\partial^2 \hat{u}}{\partial x_1^2} + \frac{\partial^2 \hat{u}}{\partial x_2^2} + k^2 \hat{u} - q(x_{1,i}, x_{2,i}) \right)^2, \tag{30}$$

where:

- $N_f$: Number of collocation points sampled from the interior of the domain $\Omega$,
- $\hat{u}$: Predicted solution obtained from the neural network,
- $q(x_{1,i}, x_{2,i})$: Source term evaluated at the collocation point $(x_{1,i}, x_{2,i})$.

**Boundary Condition Loss**  The boundary loss enforces zero displacement at the edges of the membrane, consistent with the fixed boundary condition:

$$\mathcal{L}_{\text{boundary}} = \frac{1}{N_b} \sum_{i=1}^{N_b} \hat{u}(x_{1,i}, x_{2,i})^2, \tag{31}$$

where $N_b$ is the number of points sampled from the boundary $\partial\Omega$.

**Total Loss**  The total loss combines the residual and boundary condition losses:

$$\mathcal{L}(\theta) = \lambda_r \mathcal{L}_{\text{residual}} + \lambda_b \mathcal{L}_{\text{boundary}}, \tag{32}$$

where $\lambda_r$ and $\lambda_b$ are weighting coefficients balancing the contributions of the residual and boundary losses.

### B.1.3 2D INCOMPRESSIBLE NAVIER-STOKES (WITH LEARNABLE COEFFICIENTS)

We consider the two-dimensional incompressible Navier-Stokes equations in non-dimensional form, augmented with learnable coefficients for the advection and viscosity terms:

$$u_t + \lambda_{\text{adv}}(u u_x + v u_y) = -p_x + \lambda_{\text{vis}}(u_{xx} + u_{yy}), \tag{33}$$
$$v_t + \lambda_{\text{adv}}(u v_x + v v_y) = -p_y + \lambda_{\text{vis}}(v_{xx} + v_{yy}), \tag{34}$$
$$u_x + v_y = 0. \tag{35}$$

Here, $u(t, x, y)$ and $v(t, x, y)$ are the velocity components, $p(t, x, y)$ is the pressure, and $\lambda_{\text{adv}}, \lambda_{\text{vis}}$ are trainable scalars. In a *forward* setting, $p$ is predicted by the network but not supervised; in an *inverse* setting, $\lambda_{\text{adv}}$ and $\lambda_{\text{vis}}$ are learned from data.

**Network Outputs**  Let a neural network $\mathcal{N}_\theta(t, x, y)$ predict

$$(\hat{u}, \hat{v}, \hat{p}) = \mathcal{N}_\theta(t, x, y),$$

**Residual Loss**  Define the PDE residuals at collocation points $\{(t_i, x_i, y_i)\}_{i=1}^{N_f}$ by

$$f_u = \hat{u}_t + \lambda_{\text{adv}}(\hat{u}\,\hat{u}_x + \hat{v}\,\hat{u}_y) + \hat{p}_x - \lambda_{\text{vis}}(\hat{u}_{xx} + \hat{u}_{yy}), \tag{36}$$
$$f_v = \hat{v}_t + \lambda_{\text{adv}}(\hat{u}\,\hat{v}_x + \hat{v}\,\hat{v}_y) + \hat{p}_y - \lambda_{\text{vis}}(\hat{v}_{xx} + \hat{v}_{yy}), \tag{37}$$
$$f_c = \hat{u}_x + \hat{v}_y, \tag{38}$$

and the residual loss

$$\mathcal{L}_r = \frac{1}{N_f} \sum_{i=1}^{N_f} \Big( |f_u(t_i, x_i, y_i)|^2 + |f_v(t_i, x_i, y_i)|^2 + \alpha\, |f_c(t_i, x_i, y_i)|^2 \Big), \tag{39}$$

where $\alpha > 0$ weights the continuity residual.

**Data Loss (Vortex Shedding Velocities)**  Given velocity measurements from the vortex-shedding flow, $\{(t_i, x_i, y_i, u_i, v_i)\}_{i=1}^{N_d}$, we use only velocity supervision:

$$\mathcal{L}_d = \frac{1}{N_d} \sum_{i=1}^{N_d} \Big( (\hat{u}(t_i, x_i, y_i) - u_i)^2 + (\hat{v}(t_i, x_i, y_i) - v_i)^2 \Big). \tag{40}$$

Pressure is *not* supervised and is recovered implicitly via the momentum equations.

**Total Loss**  The training objective combines residual and data terms:

$$\mathcal{L}(\theta, \lambda_{\mathrm{adv}}, \lambda_{\mathrm{vis}}) \;=\; \lambda_r \, \mathcal{L}_r \;+\; \lambda_d \, \mathcal{L}_d, \tag{41}$$

with weights $\lambda_r, \lambda_d > 0$. For inverse problems, $\lambda_{\mathrm{adv}}$ and $\lambda_{\mathrm{vis}}$ are optimized jointly with $\theta$; for forward problems they are fixed (e.g., $\lambda_{\mathrm{adv}} = 1$, $\lambda_{\mathrm{vis}} = 1/\mathrm{Re}$).

## C   HESSIAN-BASED LOSS-LANDSCAPE VISUALIZATION

To better understand the local geometry of the trained PINN, we analyze the curvature of each loss component—total loss, data loss, and residual loss—around the converged weights $\theta^*$. Concretely:

1. **Get our losses.** Let
$$L_{\mathrm{tot}}(\theta), \quad L_{\mathrm{data}}(\theta), \quad L_{\mathrm{res}}(\theta)$$
   denote the total, data, and residual losses, respectively, as in Eq. (2).

2. **Compute Hessians.** For each loss $L \in \{L_{\mathrm{tot}}, L_{\mathrm{data}}, L_{\mathrm{res}}\}$, form the Hessian
$$H_L = \nabla_\theta^2 L(\theta)\big|_{\theta=\theta^*} \, .$$

3. **Eigen-decomposition.** Solve
$$H_L \, v_i = \lambda_i \, v_i, \quad \lambda_1 \geq \lambda_2 \geq \ldots,$$
   and retain the top two eigenpairs $(\lambda_1, v_1)$, $(\lambda_2, v_2)$, which capture the directions of greatest curvature.

4. **Parameter perturbation.** For offsets $(\alpha, \beta) \in [-\delta, \delta]^2$, define
$$\theta(\alpha, \beta) \;=\; \theta^* \;+\; \alpha \, v_1 \;+\; \beta \, v_2 \, .$$

5. **Landscape slice.** The two-dimensional landscape
$$f_L(\alpha, \beta) \;=\; L\big(\theta(\alpha, \beta)\big)$$
   is evaluated on a uniform grid of $(\alpha, \beta)$ and visualized as a heatmap.

**2D Helmholtz Loss-Landscape Visualization.** In 7a we see that in case that model has converge to a trivial solution that has trapped in steep 1 dimension wells and stops which does nto faciliates learning. on the other hand 7b we that a converged model has smoother optimization bowls that facilates good convergence

## D   EXPERIMENTAL SETUP

### D.1   DATASET DESCRIPTION

**1D Convection.** We consider the periodic 1D Convection equation (Krishnapriyan et al., 2021)

$$\frac{\partial u}{\partial t} + \beta \, \frac{\partial u}{\partial x} = 0, \quad x \in [0, 2\pi], \; t \in [0, T], \tag{42}$$

with

$$u(x, 0) = \sin(x), \quad u(0, t) = u(2\pi, t). \tag{43}$$

Here $\beta > 0$ is the *convection coefficient*—larger $\beta$ yields faster transport and sharper solution gradients, increasing PDE difficulty. The mode-wise analytic solution is

$$u_{\mathrm{a}}(x, t) = \mathcal{F}^{-1}\big[\mathcal{F}(\sin x) \, e^{-i\beta k \, t}\big]. \tag{44}$$

**2D Inhomogeneous Helmholtz.** On the spatial domain $\Omega = [-1, 1]^2$, we consider the steady-state Helmholtz equation with inhomogeneous Dirichlet boundary conditions (Wang et al., 2020):

$$\nabla^2 u + k^2 \, u = q(x_1, x_2), \quad u\big|_{\partial\Omega} = 0, \tag{45}$$

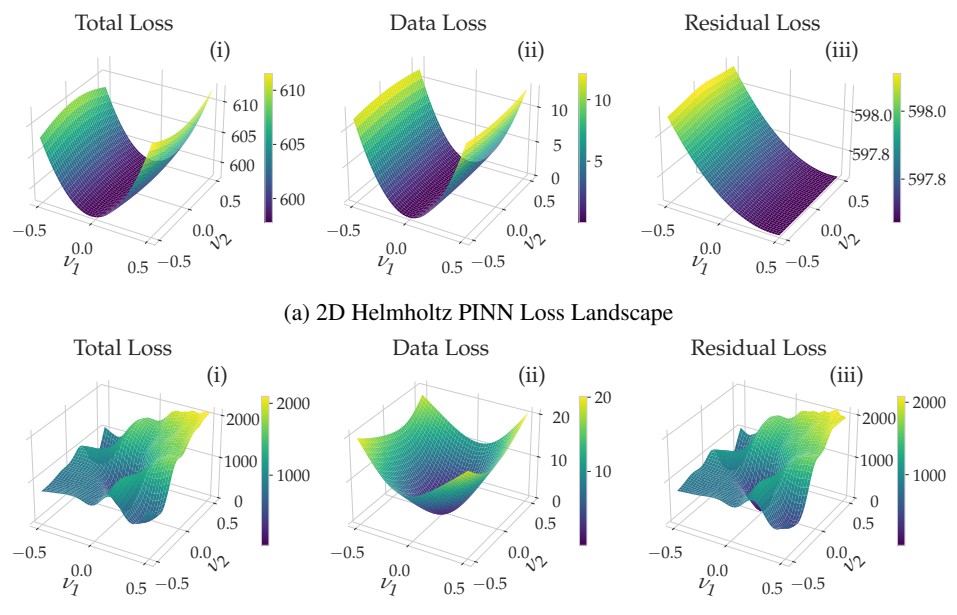

(a) 2D Helmholtz PINN Loss Landscape

(b) 2D Helmholtz LeIn-PINN Loss Landscape

Figure 7: Loss Landscape visualization by pertubating model along its top-2 eigenvectors ($\nu_1$,$\nu_2$) plots for PINN Training (Total, Data, Residuals) in the two case of PINN in 2D Helmholtz System($a_{55}$): (a) Randomly Initialized PINN, (b) LeIn-PINN.

where $k > 0$ denotes the wavenumber, and integer mode numbers $a_1, a_2 \in \mathbb{Z}^+$ determine the spatial oscillation patterns and the overall complexity of the PDE system. We select the analytical solution and corresponding source term:

$$u_{\mathrm{a}}(x_1, x_2) = \sin(a_1\pi x_1)\,\sin(a_2\pi x_2), \tag{46}$$

$$q(x_1, x_2) = \left(k^2 - (a_1\pi)^2 - (a_2\pi)^2\right)\,\sin(a_1\pi x_1)\,\sin(a_2\pi x_2) \tag{47}$$

with a fixed wavenumber $k = 1$. Note, we specifically choose $k^2 \neq a_1^2 + a_2^2$ so that the selected $k$ does not correspond to an eigenmode of the domain, resulting in a non-zero source term $q(x_1, x_2)$. Consequently, $a_1$ and $a_2$ serve as the primary PDE parameters affecting solution complexity and difficulty.

### D.2 BASELINE MODEL DESCRIPTION

We now provide brief descriptions of baseline models employed for comparative evaluation.

1. PINN (fixed) Raissi Original Formulation

2. Curr-Reg. Krishnapriyan et al. (2021): Curriculum regularization of PINN which progressively trains models from easier to harder PDE tasks by incrementally introducing complexity during training. If we want to Learn a 1D convection system with $\beta = 30$. We incremently train on $\beta = \{5, 10, 15, 20, 25, 30\}$

3. PINN (dynamic) Wang et al. (2021a); Wu et al. (2023): The current optimized and stable PINN training paradigm. In each iteration, collocation points are randomly resampled using quasi-random low-discrepancy sequences, combined with adaptive learning rate scheduling to improve convergence.

4. R3 Daw et al. (2023): Employs the adaptive sampling strategy Retain-Resample-Release (R3), focusing training effort by retaining collocation points with high residual errors while periodically resampling the remaining points.

5. Meta-PDE  Qin et al. (2022): A meta-learning framework for PDEs based on a MAML-style formulation. It also includes a learnable per-parameter step-size.

6. LeIn-PINN w/o (IE, GLO): Effectively standard PINN training from randomly initialized weights.

7. LeIn-PINN w/o GLO: An adaptation of the Model-Agnostic Meta-Learning–based framework.

### D.3 EXPERIMENT SETTINGS - PDE SYSTEMS

We summarize the experimental configurations used in our study. Table 5 reports the settings for the 1D Convection system, Table 6 reports the settings for the 2D Helmholtz system, and Table 7 reports the settings for the 2D Navier-Stokes system.

Table 5: 1D Convection Experimental Settings

| Setting | Value |
| --- | --- |
| Equation | $\partial_t u + \beta\, \partial_x u = 0$ |
| Input/Domain | $(x, t) \in [0, 2\pi] \times [0, 1]$ |
| Network Architecture | 5 hidden layers, 50 neurons each, `Tanh`, Xavier init. |
| Optimizer | Adam, lr $= 0.005$ (cosine annealing) |
| Epochs | 50,000 |
| Collocation Points | $N_f = 1000, \quad$ Boundary Points: $N_b = 1000$ |
| Training Tasks | $\beta \in \{5, 10, 15, 20, 25\}$ |
| Evaluation Tasks (Interp.) | $\beta \in \{7.5, 12.5, 17.5, 22.5\}$ |
| Evaluation Tasks (Extra.) | $\beta \in \{30, 40, 50, 60, 70, 80\}$ |

Table 6: 2D Helmholtz Experimental Settings

| Setting | Value |
| --- | --- |
| Equation | $\Delta u + k^2 u = q(x_1, x_2),\ \ k = 1$ |
| Input/Domain | $(x_1, x_2) \in [-1, 1]^2$ |
| Sampling | Halton sequence in $[-1, 1]^2$ |
| Network Architecture | 5 hidden layers, 50 neurons each, `Tanh`, Xavier init. |
| Optimizer | Adam, lr $= 0.005$ (cosine annealing) |
| Epochs | 50,000 |
| Collocation Points | $N_f = 1000, \quad$ Boundary Points: $N_b = 1000$ |
| Training Tasks | $(a_1, a_2) \in \{(1, 1), (1, 2), (1, 3), (1, 4), (1, 5)\}$ |
| Evaluation Tasks (Interp.) | $(a_1, a_2) \in \{(4, 4), (4, 5), (5, 5)\}$ |
| Evaluation Tasks (Extra.) | $(a_1, a_2) \in \{(4, 6), (5, 6), (6, 6)\}$ |

## E    ADDITIONAL RESULTS AND ABLATIONS

In this section, we present supplementary training results to further evaluate the behavior of our methods.

### E.1    NAVIER-STOKES

As a qualitative stress test in a multi-physics regime, we extend our analysis to the 2D incompressible, time-dependent Navier-Stokes cylinder wake at Reynolds number $\mathrm{Re} = 1000$. We consider both

Table 7: 2D Navier-Stokes Experimental Settings (cylinder wake)

| Setting | Value |
| --- | --- |
| Equation | Incompressible NS with learnable $\lambda_{\text{adv}}$, $\lambda_{\text{vis}}$ |
| Input/Domain | $(x, y, t) \in \Omega \times [0, T]$; $\Omega$ is channel with unit-diameter cylinder |
| Network Architecture | 7 hidden layers, 50 neurons each, `Tanh`, Xavier init. |
| Optimizer | Adam, lr $= 0.005$ (cosine annealing) |
| Epochs | 50,000 |
| Collocation Points | $N_f = 5000$ |
| Training Tasks (Re) | $\text{Re} \in \{100, 200, 300, 400, 500\}$ |
| Evaluation Tasks | $\text{Re} \in \{600, 800, 1000\}$ |

forward and inverse PINN formulations; in the inverse setting, the advection and viscosity coefficients are treated as unknowns and learned jointly as trainable parameters.

In Fig. 2a, we show the simulation setup with vortex shedding behind a cylinder, highlighting the subdomain used for training. Figs. 2b and 2c display representative velocity components $(u_x, u_y)$ at a chosen time slice, which form part of the forward prediction targets. Turning to pressure reconstruction, the ground-truth field at $\text{Re} = 1000$ is shown in Fig. 2d. A randomly initialized PINN (Fig. 2e) fails to capture the complex wake structures and exhibits a clear failure case, whereas LeIn-PINN (Fig. 2f) produces a qualitatively faithful reconstruction of the pressure distribution.

In the inverse formulation, where the PDE coefficients are unknown and must be inferred from sparse data, we again observe that vanilla PINNs face significant challenges, while LeIn-PINN demonstrates improved stability and accuracy. Together, these results indicate that the benefits of learned initialization extend beyond canonical PDEs to realistic, time-dependent fluid dynamics governed by Navier-Stokes.

### E.2 INDIVIDUAL TRAINING LOSS AND ERROR

In this subsection, we examine individual training trajectories for both the randomly initialized PINN and our proposed LeIn-PINN. For each run, we plot the total, data, and residual losses alongside the mean absolute error as functions of the training epoch. Results are shown for both the 1D Convection equation and the 2D Helmholtz equation, highlighting run-to-run variability and the robustness of our initialization strategy.

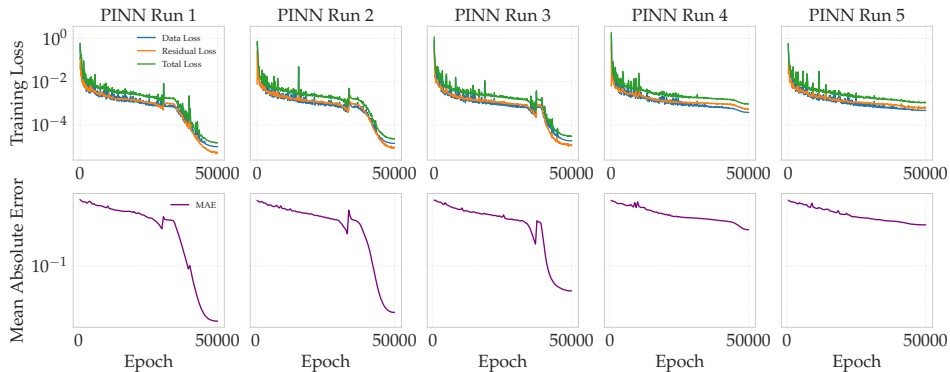

Figure 8: Individual training runs and their Training Loss and Mean absolute errors for the Random-initialized PINN on the 1D Heat Equation ($\beta = 70$).

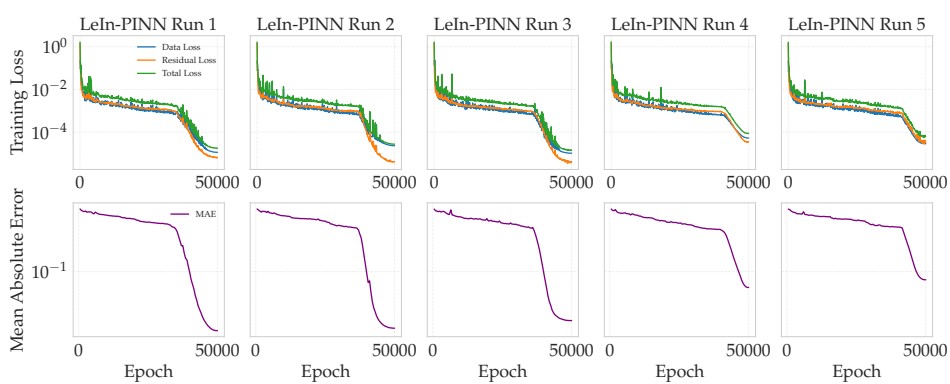

Figure 9: Individual training runs and their Training Loss and Mean absolute errors for the LeIn-PINN on the 1D Heat Equation ($\beta = 70$).

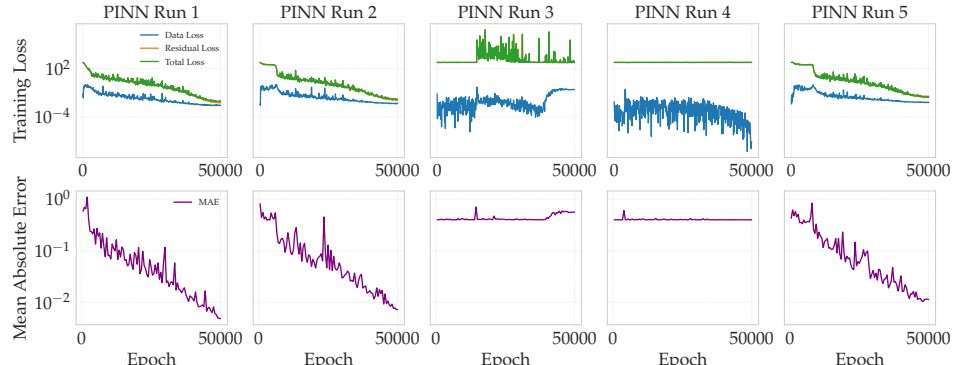

Figure 10: Individual training runs and their Training Loss and Mean absolute errors for the Random-initialized PINN on the 2D Helmholtz Equation ($a_{66}$).

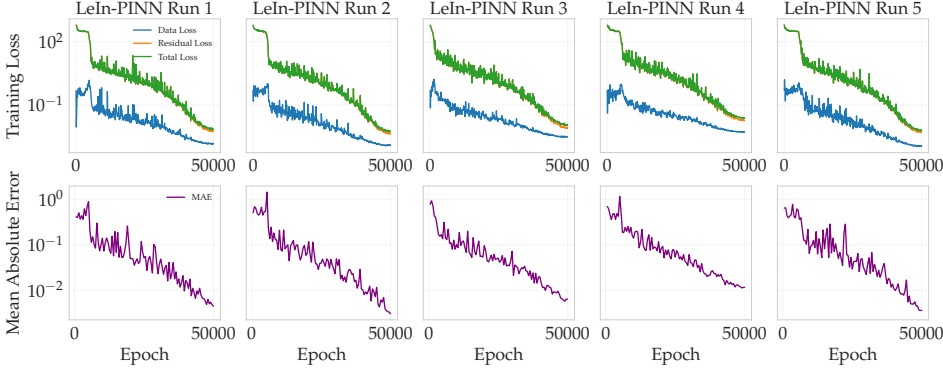

Figure 11: Individual training runs and their Training Loss and Mean absolute errors for the LeIn-PINN on the 2D Helmholtz Equation ($a_{66}$).

### E.3 (RQ3): What is the effect of invariance encoding and gated layer-wise optimization in LeIn-PINN? (Ablation Analysis)

**Mean Log-Energy Analysis.** We further examine error distribution by plotting the mean log-energy spectrum of the absolute error fields corresponding to LeIn-PINN and its ablation variants. In Fig. 12, we noticed that across both the 1D Convection and 2D Helmholtz cases, LeIn-PINN shows a markedly lower overall energy distribution compared to Random (Xavier) and MAML. This confirms that our layer-wise optimization (i.e., GLO) not only reduces average error but also suppresses energy in all frequency bands, including the challenging high-frequency components.

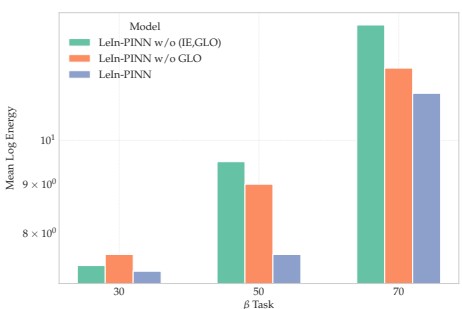 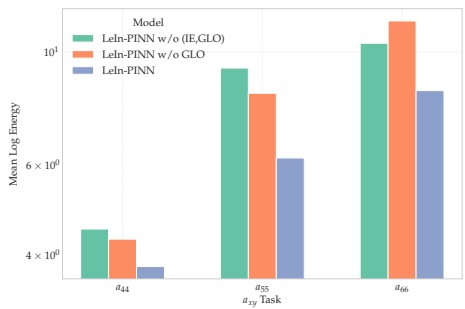

(a) 1D Convection PINN log-Energy across frequency spectrum

(b) 2D Helmholtz PINN log-Energy across frequency spectrum

Figure 12: Comparison of the mean energy density for the 1D Convection PINN (a) and the 2D Helmholtz PINN (b).

### E.4 Extended Results: Mean and Standard Deviation

We provide additional analysis here by reporting the mean and standard deviation of all experiments. Tables 8-10 summarize the full results across all extrapolation settings for the 1D convection, 2D Helmholtz systems and 2D Navier-Stokes systems respectively. These tables complement the main paper by showing the variability across independent runs and highlight the stability of LeIn-PINN compared to the state-of-the-art baselines.

Table 8: Mean Absolute Error (MAE) on 1D convection for extrapolation tasks. Values report the mean MAE across 5 random seeds, with corresponding standard deviation in parentheses.

| Task | PINN (dynamic) | Curr-Reg. | R3 | Meta-PDE | LeIn-PINN |
|------|----------------|-----------|-----|----------|-----------|
| 30 | 0.0071 (0.0023) | 0.1403 (0.1286) | **0.0067 (0.0022)** | 0.04336 (0.01674) | 0.0092 (0.0025) |
| 40 | 0.0126 (0.0051) | 0.2719 (0.1641) | 0.0678 (0.1061) | 0.1588 (0.14050) | **0.0122 (0.0045)** |
| 50 | 0.0226 (0.0072) | 0.3604 (0.1036) | 0.3224 (0.1387) | 0.2112 (0.12703) | **0.0150 (0.0055)** |
| 60 | 0.1607 (0.1922) | 0.3929 (0.0202) | 0.4319 (0.0238) | 0.3757 (0.15479) | **0.0172 (0.0046)** |
| 70 | 0.3670 (0.0267) | 0.4566 (0.0446) | 0.4726 (0.0208) | 0.35702 (0.02135) | **0.0272 (0.0103)** |

Table 9: Mean Absolute Error (MAE) on the 2D Helmholtz PDE across varying $a_{xy}$ in the extrapolation setting. Values report the mean MAE across 5 random seeds, with corresponding standard deviation reported in parentheses.

| Task | PINN (dynamic) | Curr-Reg. | R3 | Meta-PDE | LeIn-PINN |
|------|----------------|-----------|-----|----------|-----------|
| $a_{44}$ | 0.0047 (0.0050) | 0.4131 (0.0264) | 0.3864 (0.0360) | 0.2473 (0.1541) | **0.0013 (0.0004)** |
| $a_{45}$ | 0.2154 (0.3049) | 0.6172 (0.2846) | 1.5196 (2.4908) | 0.4736 (0.3218) | **0.0029 (0.0006)** |
| $a_{55}$ | 0.1962 (0.2643) | 0.4106 (0.0105) | 0.4173 (0.0215) | 0.5982 (0.3420) | **0.0058 (0.0035)** |
| $a_{46}$ | 0.0850 (0.1744) | 0.4050 (0.0056) | 0.4272 (0.0356) | 0.4954 (0.3495) | **0.0050 (0.0009)** |
| $a_{56}$ | 0.0879 (0.1728) | 0.4390 (0.0683) | 0.4139 (0.0143) | 0.4077 (0.0092) | **0.0081 (0.0015)** |
| $a_{66}$ | 0.1687 (0.2153) | 0.4346 (0.0685) | 0.4204 (0.0197) | 0.4329 (0.0578) | **0.0150 (0.0107)** |

Table 10: Pressure error (MAE) for 2D Navier–Stokes across Reynolds numbers. Values are mean with standard deviation in parentheses.

| Re | PINN (dynamic) | Curr-Reg. | R3 | Meta-PDE | LeIn-PINN |
|---|---|---|---|---|---|
| 600 | 0.2486 (0.3670) | 0.2133 (0.1783) | 0.2456 (0.3938) | 0.3172 (0.0400) | 0.0057 (0.0005) |
| 800 | 0.0105 (0.0057) | 0.2150 (0.1753) | 0.0137 (0.0066) | 0.2997 (0.0262) | 0.0058 (0.0009) |
| 1000 | 0.0243 (0.0337) | 0.2147 (0.1757) | 0.0426 (0.0428) | 0.2812 (0.0342) | 0.0057 (0.0001) |

### E.5 ADDITIONAL RESULTS: EFFECT OF DIFFERENT INITIALIZATION

Table 11 compares the extrapolation MAE for PINN-Dynamic under Xavier initialization, PINN with Kaiming initialization, and PINN-Dynamic with LeIn (i.e., nothing but our proposed LeIn-PINN model). The learned initialization (i.e., PINN-Dynamic LeIn) achieves consistently lower error than both Xavier and Kaiming baselines as task get progressively harder, indicating that the learned invariance weights provide a stronger starting point for optimization than standard random initializations.

Table 11: Error (1D convection, MAE) for extrapolation tasks. Values show mean with standard deviation in parentheses.

| Task | PINN (dynamic) Xavier | PINN (dynamic) Kaiming | PINN (dynamic) LeIn |
|---|---|---|---|
| 30 | **0.0071 (0.0023)** | 0.0092 (0.00256) | 0.0092 (0.0025) |
| 40 | 0.0126 (0.0051) | 0.01582 (0.00288) | **0.0122 (0.0045)** |
| 50 | 0.0226 (0.0072) | 0.01762 (0.00232) | **0.0150 (0.0055)** |
| 60 | 0.1607 (0.1922) | 0.02098 (0.00572) | **0.0172 (0.0046)** |
| 70 | 0.3670 (0.0267) | 0.07076 (0.06444) | **0.0272 (0.0103)** |

Table 12 compares our LeIn-PINN with the parameterized PINN method (P$^2$INN). In the 1D convection experiments, trained on $\beta = [5, 10, 15, 20, 25]$, P$^2$INN fails to converge on the challenging extrapolation tasks, whereas LeIn-PINN successfully learns the target solutions.

Table 12: Error (2D abs) on extrapolation tasks for and P$^2$INN and LeIn-PINN,

| Task | P2INN | LeIn-PINN |
|---|---|---|
| 30 | 0.2259 (0.0202) | **0.0092 (0.0025)** |
| 40 | 0.2485 (0.0091) | **0.0122 (0.0045)** |
| 50 | 0.2561 (0.0023) | **0.0150 (0.0055)** |
| 60 | 0.2559 (0.0014) | **0.0172 (0.0046)** |
| 70 | 0.2579 (0.0009) | **0.0272 (0.0103)** |

**Layer-wise Parameter Distribution Analysis.** To characterize how the learned initialization modifies the PINN parameter landscape, we compare the layer-wise weight distributions of LeIn-PINN against layer-wise weight distributions of a randomly initialized PINN. We report (Table. 13) two statistics for the first, middle, and final layers across all PDEs namely: (i) the absolute difference between the variance of weight distributions per layer $\Delta = |\text{Var}(\text{LeIn-PINN}) - \text{Var}(\text{Random PINN})|$, and (ii) the 1-Wasserstein distance, which measures distributional similarity.

Table 13: Layer-wise statistics across PDEs. Variance $\Delta$ captures the absolute scale difference and Wasserstein distance quantifies distributional similarity.

| Layer | 1D Convection | | 2D Helmholtz | | 2D Navier–Stokes | |
|---|---|---|---|---|---|---|
| | Variance $\Delta$ | Wasserstein | Variance $\Delta$ | Wasserstein | Variance $\Delta$ | Wasserstein |
| First Layer | 0.058103 | 0.108291 | 0.013771 | 0.038730 | 0.004854 | 0.039123 |
| Middle Layer | 0.009142 | 0.024560 | 0.002109 | 0.006187 | 0.001054 | 0.004970 |
| Last Layer | 0.000264 | 0.002663 | 0.000123 | 0.002950 | 0.000459 | 0.004678 |

In Table 13, we find an interesting behavior where the results show a consistent pattern across systems. Specifically, we found a stark distributional dissimilarity (between randomly initialized PINNs and PINNs initialized with learned weights) in shallower layers with increased distributional similarity in deeper layers. This difference in distributional similarity can be attributed to a higher variance of weights in shallower layers of LeIn-PINN, compared to variance of weights in the corresponding layers of randomly initialized PINNs. Previous work has shown that neural networks with higher variance of weights in shallower layers have increased expressivity.Yang & Schoenholz (2017), thereby allowing us to infer that LeIn-PINN with learned weights offer greater expressive power (and thereby larger learning capacity) compared to randomly initialized PINNs, also contributing to their superior performance in challenging PDE domains.

### E.6 ADDITIONAL RESULTS: COMPATIBILITY WITH STATE-OF-THE-ART METHODS

We would like to highlight that our proposed method is *model-agnostic* and fully compatible with other PINN variants such as the R3 sampling model (R3) Daw et al. (2023). To evaluate this, we compare the performance of two variants of the R3 sampling model:

- **R3 (Learned Init.)**: initialized with LeIn weights obtained from our invariance encoding setup,
- **R3 (Xavier Init.)**: initialized with standard Xavier initialization.
- **R3 (Kaiming Init.)**: initialized with standard Kaiming initialization.

Each variant was trained and evaluated on four challenging 1D Convection PDE contexts with $\beta \in \{30, 40, 50, 60, 70\}$. Performance is reported using the mean absolute error (MAE). As shown in Table 14, initialization with LeIn consistently reduces MAE across all tasks compared to Xavier initialization, confirming that our approach improves state-of-the-art models.

Table 14: Comparison of R3 model performance on 1D Convection system with Xavier vs. Learned Initialization. Results reported as mean absolute error (MAE).

| 1D Convection ($\beta$) | R3 (Xavier Init.) | R3 (Learned Init.) | MAE Reduction (%) |
|---|---|---|---|
| 30 | 0.0067 | 0.0055 | 17.91% |
| 40 | 0.0678 | 0.0106 | 84.37% |
| 50 | 0.3224 | 0.1050 | 67.43% |
| 60 | 0.4319 | 0.3670 | 15.02% |
| 70 | 0.4726 | 0.3384 | 28.39% |

### E.7 ADDITIONAL RESULTS: SENSITIVITY ANALYSIS OF LEIN-PINN TO NUMBER AND RANGE OF IE TASKS

We further analyze the sensitivity of LeIn-PINN to two factors during the Invariance Encoding (IE) phase: (i) the number of sampled tasks $k$, and (ii) the range of tasks used. For fairness, the baseline task ranges in the main paper were selected to align with prior work Krishnapriyan et al. (2021), which we also compared against (i.e., Curr-Reg. baseline).

**Sensitivity to Number of Tasks $k$.** We varied $k \in \{2, 3, 4\}$ during meta-training (note that $k = 3$ was used in the main paper). Evaluation focused on the most challenging extrapolation regimes of the 1D convection PDE ($\beta = 50, 60, 70, 80$), with results averaged over 5 random seeds. As shown in Table 15, performance is relatively stable across different $k$. In practice, we recommend setting $k$ to 60-80% of the available tasks. If further tuning is needed, $k$ may be selected via a validation set by sweeping values between 50-70%.

**Sensitivity to Task Range.** We further test the robustness of LeIn-PINN by varying the range of easy tasks selected for IE. In addition to the main setup LeIn-PINN(5-25), we evaluated LeIn-PINN(10-30) and LeIn-PINN(15-35). The evaluation tasks remained the challenging extrapolation regimes

Table 15: Sensitivity of LeIn-PINN to the number of sampled tasks $k$ during IE phase. Reported as mean absolute error (MAE).

| 1D Convection ($\beta$) | $k = 2$ | $k = 3$ | $k = 4$ |
|---|---|---|---|
| 50 | 0.0168 | 0.0150 | 0.01674 |
| 60 | 0.0210 | 0.0172 | 0.02978 |
| 70 | 0.0529 | 0.0272 | 0.02724 |
| 80 | 0.3878 | 0.2490 | 0.26194 |

($\beta = \{50, 60, 70, 80\}$). Results in Table 16 show that all LeIn-PINN variants achieve comparable performance, with only minor differences on the hardest case ($\beta = 80$). Most importantly, all LeIn-PINN variants tested (irrespective of the task range chose for IE initialization), consistently outperform state-of-the-art baselines, including R3 Krishnapriyan et al. (2021), PINN-Dynamic Wang et al. (2020).

Table 16: Sensitivity of LeIn-PINN to task range during IE phase. Mean absolute error (MAE) is reported for $\beta \in \{50, 60, 70, 80\}$ in 1D Convection.

| 1D Convection ($\beta$) | PINN-Dynamic | R3 (R3) | LeIn (5-25) | LeIn (10-30) | LeIn (15-35) |
|---|---|---|---|---|---|
| 50 | 0.0226 | 0.3224 | 0.0150 | 0.0102 | 0.0115 |
| 60 | 0.1607 | 0.4319 | 0.0172 | 0.0204 | 0.0215 |
| 70 | 0.3670 | 0.4726 | 0.0272 | 0.0362 | 0.0964 |
| 80 | 0.4634 | 0.4988 | 0.2490 | 0.0583 | 0.1397 |

**Summary.** These results suggest that LeIn-PINN is robust to both (i) the number of sampled tasks $k$ and (ii) the range of tasks selected during the IE phase. Domain knowledge about PDE dynamics can be effectively leveraged to design task ranges for IE without sacrificing performance. Further, regardless of the value of $k$ or the easy-task range, LeIn-PINN consistently outperforms state-of-the-art randomly initialized PINN variants on challenging PDE regimes.

### E.8 ADDITIONAL RESULTS: IMPACT OF EXTENDED TRAINING DURATION

In our experience, convergence failures of PINNs on challenging PDE domains are more dependent on *effective training dynamics* rather than simply extending training steps. To address this directly, we conducted a controlled experiment on the 1D Convection PDE to contrast the effect of intelligent initialization (LeIn-PINN) with that of longer training.

Specifically, we ensured that all methods underwent the same total number of gradient update steps (56K). In our setup, LeIn-PINN was trained with 6K meta-training steps to learn the initialization, followed by 50K fine-tuning steps on the target task. In contrast, Curr-Reg. and R3 and baselines were trained for 56K steps in total, matching the gradient update budget of LeIn-PINN.

Table 17 shows that there remains a *significant performance gap* between LeIn-PINNs and the baseline models across all extrapolation regimes. This confirms that simply increasing training duration is insufficient to address PINN failures; in contrast, intelligent initialization plays a crucial role.

Table 17: Effect of extended training duration on 1D Convection system. All models are trained for 56K total gradient steps. Reported as MAE.

| 1D Convection ($\beta$) | Curr-Reg.(56K) | R3(56K) | LeIn-PINN |
|---|---|---|---|
| 50 | 0.3004 | 0.3291 | 0.0150 |
| 60 | 0.4015 | 0.4356 | 0.0172 |
| 70 | 0.4208 | 0.4565 | 0.0272 |
| 80 | 0.4738 | 0.4648 | 0.2490 |

### E.9 COMPUTATIONAL RESOURCE

All experiments were performed on server with a RTX A6000 GPUs with 48GB vRAM.

To ensure reproducibility, we document the hardware and software environments used, as well as the approximate runtimes for our main experiments.

Table 18: Hardware and Software Specifications

| Component | Specification |
|---|---|
| GPU | NVIDIA RTX A6000 (48 GB VRAM) |
| CPU | Intel(R) Xeon(R) Platinum 8358 CPU @ 2.60GHz |
| System RAM | 2 TB DDR4 |
| Operating System | Ubuntu 20.04 LTS |
| Python | 3.10 |
| PyTorch | 2.4 (CUDA 12) |
| Experiment Tracking | MLflow 2.14 |

**Computation Cost.** To provide a holistic context regarding the compute cost of the initialization strategy, we report wall-clock training time (in minutes) and peak GPU memory usage (in MB) in Table 19. All measurements were conducted on an NVIDIA RTX A6000 GPU (48 GB vRAM) to ensure consistency Peak memory usage was measured using `nvidia-smi`.

Table 19: Computation cost for 1D Convection system. We report wall-clock training time (minutes) and peak memory (MB).

| Stage / Method | Time (min) | Peak GPU Memory (MB) |
|---|---|---|
| LeIn-PINN (Invariance Encoding) | 3.1 | 740 |
| LeIn-PINN (Fine-Tuning) | 10.4 | 540 |
| Xavier Init. PINN Training | 10.4 | 540 |

We emphasize that the meta-training stage (invariance encoding) is a *one-time cost*. Once the initialization is learned, it can be reused across multiple PDE tasks in the same domain class, thereby amortizing this overhead. Furthermore, the collocation points and grid resolution were kept constant between the invariance encoding and task-specific fine-tuning stages. These results confirm that LeIn-PINN scales comparably to standard PINNs in terms of grid resolution, with the only additional overhead being the upfront, one-time meta-training stage.

### ACKNOWLEDGMENT

We acknowledge the use of a large language model (LLM) to check spelling and grammar in this manuscript.

