# OpenReview forum: "LeIn-PINN: Learned Initialization to Alleviate Convergence Failures in Physics Informed Neural Networks"
_ICLR.cc/2026/Conference — Submitted to ICLR 2026_

### Official Review · Reviewer_26a7 · 2025-10-27

**Soundness:** 2
**Presentation:** 2
**Contribution:** 2
**Rating:** 4
**Confidence:** 4

**Summary:**

The paper introduces the use of a learned NN initialisation to improve the PINN training convergence. This is done by meta-learning-like approach, with a gating mechanism to focus the learned initialisation to some parameters at a time. The results show improved generalisation accuracy compared to other methods that work on other aspects of PINN training.

**Strengths:**

The paper attempts to solve the issue of PINN training convergence via the PINN initialisation. This viewpoint is not too common, which is good, although have been looked at in prior works already as well (see Weaknesses section).

The experiments section is organised well. There are quite clear research questions, and some of the points were answered well and provided evidence from experiments to back it up as well.

Experiments do show promising results, such as for advection equation for high values of beta (which is a classic sanity check for PINN training).

**Weaknesses:**

Use of learned initialisation is not too well-motivated. While it may be true that changing initialisation can improve things, why can I not spend this effort into tuning other aspects of PINN training? I suspect there may be some motivation related to amortising the PINN training (there might be some reason that you have to train multiple PINNs on the same initialisation), which can make it a better case that the initialisation is the thing that we should try to fix. I think this part can strengthen the motivation of the problem by quite a bit.

Meta-learning for PINNs and learning of some PINN initialisation are already studied in the context of PINNs [1-4], as cited by authors and as other papers have that were not mentioned in the paper. It would be good to more explicitly discuss the shortcomings of these existing works with respect to improved PINN training.

Related to above, more explicitly compare the proposed methods against these benchmarks would also be useful (particularly in some of the settings that are also used in those papers). The existing methods that LeIn compares with are quite orthogonal to proposed methods (in that I could use those methods alongside with LeIn as well), so more direct comparisons with more similar methods may be more informative. If the authors compare against with just these methods, then there are also other methods for improving PINN training convergence that the authors could have chosen to compare with (regarding collocation point selection, loss function balancing, PINN architecture, etc.) which are relatively newer than the benchmarks used.

None of the reported results have error bars.

While the results show a sample of different PDE cases that trains from the learned initialisation, I feel it could be trained more extensively as well, in that a much denser set of PDE conditions could have been tried (e.g., training 50 different PDE parameters on the same initialisation for demonstration instead of just 6 different PDE parameters). This would strengthen the use of a learned initialisation.

Around Line 122 -- what is consider relatively easier PDE configurations may not be trivial, and will be PDE-dependent. A more concrete definition for this (if exists) or some concrete criteria to determine this to be used in practice may be useful.

Around Line 139 -- would like to see more motivating why the gating is needed. It seems to be done a bit, but would like to see it described more, especially connected to the results that the authors show later on regarding this part.

---

[1] Meta-learning pinn loss functions. 2022.

[2] Meta-pde: Learning to solve pdes quickly without a mesh. 2022.

[3] DATS: Difficulty-Aware Task Sampler for Meta-Learning Physics-Informed Neural Networks. 2024.

[4] PIED: Physics-Informed Experimental Design for Inverse Problems. 2025.

**Questions:**

1. How is the time required to obtain the learned initialisations? Is it sufficiently more than the time required to just train the PINNs normally with the other proposed methods for a longer period of time? A similar question can be asked for other resource usages as well, including GPU memory for training.

2. It might be interesting to see what the learned initialisation looks like after they are learned but before trained on the specific PINN case. This might provide with further understanding on why the convergence overall is better.

---

> ### Author Response · Authors · 2025-11-25
> **Q1, Q2. Response to reviewer 26a7.**
>
> We thank the reviewer for their time and the insightful questions. Please find our responses below.
>
>
>
>
>
> ## `Q1. Time and Resource Cost of Learned Initialization`
>
> ### [Cost of Learned Initialization]
>
> Thank you for the question. Time and memory cost of learned initialization LeIn-PINN (Invariance Encoding + GLO) on the 1D convection domain have been included in the table below. We would like to emphasize that the pre-training stage (i.e., learned initialization) comprising the invariance encoding + GLO is a **one-time (minimal additional) cost**. Once the initialization is learned, it can be reused to fine-tune across multiple PDE tasks, thereby amortizing this overhead.
>
> This analysis has also been added to Appendix E.9 of the paper.
>
>
> ### **Time & Memory Cost**
>
> | Stage / Method                     | Time (min) | Peak GPU Memory (MB) |
> |-----------------------------------|------------|------------------------|
> | LeIn-PINN (Invariance Encoding+GLO) | 3.1        | 740                    |
> | LeIn-PINN (Fine-Tuning)           | 10.4       | 540                    |
> | Xavier Init. PINN Training        | 10.4       | 540                    |
>
>
> ---
>
> ### [Can Training Baselines Longer Close the Gap?]
>
> To address the reviewer’s question about whether simply training competing methods longer would close the performance gap, we conducted an experiment where, on the 1D convection domain, we trained the competing methods **Curr-Reg** and **R3** for the **exact same number of epochs** as:
>
> - LeIn-PINN (Invariance Encoding+GLO): **6,000 epochs**
> - LeIn-PINN (Fine-Tuning): **50,000 epochs**
>
> Total = **56,000 epochs**.
>
> Since all methods use the same architecture, collocation points, and batch sizes, training for the same epoch count incurs roughly identical time and memory cost.
>
> Despite this equalized training budget, the results show that **LeIn-PINN remains dramatically superior**, indicating that performance improvements are **not** due to longer training time.
>
>
> ### **1D Convection (β): Equalized Training Budget (56k epochs)**
>
> | Task | Curr-Reg (56k) | R3 (56k) | LeIn-PINN |
> |------|----------------|----------|------------|
> | 50   | 0.3004         | 0.3291   | **0.0150** |
> | 60   | 0.4015         | 0.4356   | **0.0172** |
> | 70   | 0.4208         | 0.4565   | **0.0272** |
> | 80   | 0.4738         | 0.4648   | **0.2490** |
>
>
> This analysis has been added to Appendix E.8 of the paper.
>
>
>
> ## `Q2. Further Understanding the Effect of Learned Initialization`
>
> Thank you for the interesting question. To address this question, we contrast the layer-wise parameter distributions of randomly initialized PINNs and PINNs initialized with learned weights.
>
> Specifically, we report quantitative results of aggregate statistics for the first, middle, and final layers. We report two statistics:
>
> 1. **Var** Δ = |Variance(LeIn-PINN Weights) – Variance(Random PINN Weights)|
> 2. **Wasserstein distance**
>
> As reported in (Appendix E.5, Table 13), we observe a consistent behavior across systems: the first layer shows a stark distributional dissimilarity, this mismatch decreases in deeper layers, and the final-layer distributions are almost identical. This pattern arises from the higher variance of weights in the shallower layers of our learned initialization compared to those of randomly initialized PINNs. Prior work [1] has shown that higher variance in early layers corresponds to increased expressivity, which allows us to infer that **learned weights provide greater expressive power in the initial layers**. This distributional difference between layer weights across LeIn-PINNs and randomly initialized PINNs, explains why the learned initialization is able to support improved learning capacity in challenging PDE domains.
>
> These results are included in Table 13 of Appendix E.5 (and reproduced below for convenience).
>
> ### **Layer-wise Parameter Distribution Comparison**
>
> | Layer        | 1D Convection (Var Δ) | 1D (Wasserstein) | 2D Helmholtz (Var Δ) | 2D (Wasserstein) | 2D Navier–Stokes (Var Δ) | 2D NS (Wasserstein) |
> |--------------|-----------------------|-------------------|------------------------|-------------------|---------------------------|----------------------|
> | First Layer  | 0.058103              | 0.108291          | 0.013771               | 0.038730          | 0.004854                  | 0.039123             |
> | Middle Layer | 0.009142              | 0.024560          | 0.002109               | 0.006187          | 0.001054                  | 0.004970             |
> | Last Layer   |0.000264             | 0.002663          | 0.000123              | 0.002950          |0.000459                 | 0.004678             |
>
> ### References
> 1. Yang, Ge, and Samuel Schoenholz. "Mean field residual networks: On the edge of chaos." Advances in neural information processing systems 30 (2017).

---

> > ### Comment · Reviewer_26a7 · 2025-11-26
> > **Response**
> >
> > I thank the reviewer for their additional results provided to me and to the other reviewers as well. Based on the comments, I still have reservations within the weakness as I have mentioned, mainly regarding the motivation of the method and discussions/comparisons against existing meta-learning methods for PINNS, which seems to not be addressed. I therefore would like to retain my original score for the moment.

---

> > > ### Author Response · Authors · 2025-12-03
> > > **Response to reviewer 26a7**
> > >
> > > Thank you for your response. You have specifically raised two points which we have responded to, below.
> > >
> > > ## Point 1: Request to Include a Meta-Learning-based PINN Baseline
> > > To address your first point, we have conducted experiments to evaluate the performance of Meta-PDE [1], (as requested by you), on all three PDE domains. We find that LeIn-PINNs outperform Meta-PDEs on ALL evaluated contexts and achieve an **86.7%** performance improvement across all evaluated experiments, according to the mean absolute error (MAE) metric. Like all other results, Meta-PDE results were also derived across 5 random seeded runs and are reported in Tables 1 - 4 of the main paper. Meta-PDE underperforms because, we have evaluated on much more challenging configurations on all 3 PDE domains than in other papers. **Such challenging domains require careful control of training dynamics which we are able to achieve through our proposed Gated-Layerwise-Optimization (GLO) training mechanism, in LeIn-PINN that is absent in Meta-PDE.**
> > >
> > > ## Point 2: Clarification of Motivation
> > > We have re-stated the motivation below and grounded it in the context of prior work to make more explicit, why a method like ours based on learned initialization is required and why it works to address the problem of catastrophic PINN failures in challenging PDE domains.
> > >
> > > **[Motivation of Learned Initialization]:**  The main claim of our paper is that **`learned-initialization` helps alleviate the catastrophic failures that PINNs commonly succumb to, while being trained on challenging PDE domains**. In [2] the authors have shown that conflicting and imbalanced gradients between the physics-informed and data-driven loss terms, affect training dynamics, when PINNs are trained on challenging PDE domains. Separately, research on stabilizing training dynamics in general deep learning (DL) models [3], has shown that information propagation through DL models can be improved by appropriately initializing weights. With this understanding that the distribution of “initial weights” matters and can help alleviate adverse training dynamics, we propose a mechanism for “learning” the initial weights of PINN models (specifically LeIn-PINN) to enable improved information flow when training on challenging PDE domains.
> > >
> > > **[Effect of Learned Initialization]:** Our proposed LeIn-PINN models have been shown (through numerous quantitative and qualitative comparisons) to outperform state-of-the-art baselines across three challenging PDE domains (1D Convection, 2D Helmholtz, 2D Navier-Stokes), thereby demonstrating their effectiveness in overcoming catastrophic PINN failures.
> > >
> > > **[Why it Works]:** As shown in Fig. 4b, our proposed LeIn-PINN model due to the learned initialization is able to operate on smoother (and more easily navigable) loss landscapes when being trained on challenging PDE domains, as opposed to sharp, landscapes filled with ravines, which are hard to navigate and are prevalent in randomly initialized PINNs due to the non-trivial training dynamics mentioned earlier.
> > >
> > > # References
> > > 1. Qin, Tian, et al. "Meta-pde: Learning to solve pdes quickly without a mesh." arXiv preprint arXiv:2211.01604 (2022).
> > >
> > > 2. Wang, Sifan, Yujun Teng, and Paris Perdikaris. "Understanding and mitigating gradient flow pathologies in physics-informed neural networks." SIAM Journal on Scientific Computing 43.5 (2021): A3055-A3081.
> > >
> > > 3. Schoenholz, Samuel S., et al. "Deep Information Propagation." International Conference on Learning Representations. 2017.

---

### Official Review · Reviewer_SMmg · 2025-10-31

**Soundness:** 3
**Presentation:** 3
**Contribution:** 3
**Rating:** 4
**Confidence:** 3

**Summary:**

This paper addresses the convergence and generalization of PINN in challenging scenarios. A novel PINN training methodology is proposed that 1) the model parameters are initialized by easy PDE settings and 2) a gated layer-wise optimization procedure mitigates spectral bias. Experiments have been conducted on different types of PDEs compared to various baseline models to present the improved performance on extrapolation regimes.

**Strengths:**

- The generalization of PINN on unseen PDE domains is critical, and the idea of better initialization for PINNs is important.
- The methodology is well-structured.
- The experiments were extensive on various PDE domains with a detailed explanation.

**Weaknesses:**

- Some details in the methodology need to be justified.
- Additional experiments are suggested to better present the effectiveness of the proposed method.

**Questions:**

- Section 3: The bi-level optimization in the proposed method is heavily based on model-agnostic meta-learning, with the gated layer-wise optimization implemented in the outer loop. However, why the GLO improves the “hard” settings is not clear, especially since it is not applied separately to the physics-based loss and the data-driven loss. Could the authors justify how the GLO addresses the non-trivial dynamics of different loss terms?
- Section 4.1 & 4.2: The comparison of the proposed method with baseline models is unfair. The proposed model has easy settings to obtain a better initialization, while the other models are only trained and tested on hard settings. It would be more convincing to compare the baseline models trained on the same easy settings used in the proposed model and fine-tuned on hard settings.
- Table 1 - Table 3: Could the authors provide the variance of all metric numbers? The authors should provide visualizations to justify the quantitative improvements.
- Section 4.3: The ablation study is not sufficient. The design of GLO, including the number of layers and its effect on particular loss terms, should be studied.

---

> ### Author Response · Authors · 2025-11-25
> **Q1, Q2, Q4. Response to reviewer SMmg.**
>
> We thank the reviewer for their time and the insightful questions. Please find our responses below.
>
>
> ---
>
>
> ## `Q1. Clarification on Why GLO Improves “Hard” Settings`
>
>
> As investigated in previous research [1], the imbalance between the physics-guided (i.e., residual) and data-driven losses is a common reason for catastrophic failure of PINN in challenging (i.e., “hard”) PDE domains. It is also known that this imbalance is more pronounced in shallower layers of the network than in deeper layers, owing to the effect of vanishing gradients (i.e., data-driven and residual losses have slightly different computational graphs in PINNs, since residual losses require partial derivatives w.r.t inputs, while data-driven losses do not).
>
> Due to this, the data-driven loss struggles to back-propagate to shallower layers compared to the residual loss, resulting in loss imbalance. This imbalance is exacerbated when all layers are trained jointly because multiple layer weights are updated simultaneously.
>
> Motivated by this understanding, our goal behind proposing the gated layer-wise optimization (GLO) procedure was to **simplify the training dynamics** by first forcing the network to learn representations in shallower layers before progressing to deeper layers. This achieves localized learning, simplifying backpropagation dynamics compared to jointly updating all layers of the model.
>
>
> Our GLO method can also be viewed as conceptually similar to **multi-grade deep learning** [2], which is known to stabilize training, improve convergence rate, and enhance sample efficiency in deep learning models [3].
>
>
> ### References
>
> 1. Wang, Sifan, Yujun Teng, and Paris Perdikaris. "Understanding and mitigating gradient flow pathologies in physics-informed neural networks." SIAM Journal on Scientific Computing 43.5 (2021): A3055–A3081.
>
> 2. Xu, Yuesheng. "Multi-grade deep learning." Communications on Applied Mathematics and Computation (2025): 1–52.
>
> 3. Lee, Chen-Yu, et al. "Deeply-supervised nets." Artificial Intelligence and Statistics. PMLR, 2015.
>
> ---
> ## `Q2. Fairness of Baseline Comparisons`
>
>
> The main claim of our paper is in fact that learned initialization (through our proposed invariance encoding + GLO) can alleviate the catastrophic failures in challenging PDE domains that randomly initialized PINNs usually suffer from.
>
>
> ### Baselines Have Also Seen “Easy” Domains
>
> Two of our baseline models, **Curr-Reg** [1] and **P²INN** [2] (newly added), have in fact been pre-trained on the same *easy* settings before fine-tuning on the hard setting.
>
> In the table below, we analyze prediction results on the 1D convection PDE setting and show that **LeIn-PINNs significantly outperform both baselines in ALL evaluated settings**.
>
>
> ### 1D Convection: Curr-Reg vs P²INN vs LeIn-PINN
>
> | Task | Curr-Reg             | P²INN                 | LeIn-PINN            |
> |------|-----------------------|------------------------|-----------------------|
> | 30   | 0.1403 (0.1286)       | 0.2259 (0.0202)        | **0.0092 (0.0025)**   |
> | 40   | 0.2719 (0.1641)       | 0.2485 (0.0091)        | **0.0122 (0.0045)**   |
> | 50   | 0.3604 (0.1036)       | 0.2561 (0.0023)        | **0.0150 (0.0055)**   |
> | 60   | 0.3929 (0.0202)       | 0.2559 (0.0014)        | **0.0172 (0.0046)**   |
> | 70   | 0.4566 (0.0446)       | 0.2579 (0.0009)        | **0.0272 (0.0103)**   |
>
>
> ### References
>
> 1. Krishnapriyan, Aditi, et al. "Characterizing possible failure modes in physics-informed neural networks." Advances in Neural Information Processing Systems 34 (2021): 26548–26560.
>
> 2. Park, Noseong, et al. "Parameterized physics-informed neural networks for parameterized PDEs." 41st International Conference on Machine Learning, ICML 2024. ICML, 2024.
> ---
> ## `Q4. Insufficient Ablation Study for GLO Design`
>
>
> Our main claim in the paper is that our proposed Learned-Initialization Physics-Informed Neural Network (LeIn-PINN) alleviates catastrophic failures experienced by PINNs in challenging PDE domains. In light of this claim, we have proposed a GLO-based model to demonstrate through rigorous experimentation that learned initialization outperforms random initialization across variegated PDE domains.
>
> Our ablation study (Fig. 5), where we have directly compared the performance of LeIn-PINN and the LeIn-PINN variant without GLO (“LeIn-PINN w/o GLO”), successfully demonstrates the positive effect GLO has in alleviating catastrophic PINN failures in challenging PDE domains.
>
> While we agree that more sophisticated GLO schemes might indeed improve the performance of our method, this is not the goal of our paper and is left as future work.

---

> ### Author Response · Authors · 2025-11-25
> **Q3. Response to reviewer SMmg.**
>
> ## `Q3. Request for Variance Metrics and Visualizations`
>
>
> **[Mean + Standard Deviation for All Metrics]**
> Thank you for the suggestion. We agree that reporting variance strengthens the quantitative comparisons. All our experiments were run on 5 random seeds and due to space constraints, the main paper only included mean values, but we have now added full mean and standard deviation metrics for every method in Appendix E.4 (Tables 8–10).
> In response to Reviewer `26a7`, we have also incorporated the Meta-PDE baseline [1] into these extended results for completeness.
> For convenience, we also include the summary tables below. All values report the mean absolute error (MAE) across five random seeds, with the corresponding standard deviation in parentheses.
>
>
> ---
>
> ### **Table: 1D Convection Mean Absolute Error (MAE) with Standard Deviation in parentheses.**
>
> | Task | PINN (Dynamic)       | Curr-Reg              | R3                    | Meta-PDE               | LeIn-PINN             |
> |------|-----------------------|------------------------|------------------------|-------------------------|------------------------|
> | 30   | 0.0071 (0.0023)       | 0.1403 (0.1286)        | **0.0067 (0.0022)**    | 0.04336 (0.01674)       | 0.0092 (0.0025)        |
> | 40   | 0.0126 (0.0051)       | 0.2719 (0.1641)        | 0.0678 (0.1061)        | 0.1588 (0.14050)        | **0.0122 (0.0045)**    |
> | 50   | 0.0226 (0.0072)       | 0.3604 (0.1036)        | 0.3224 (0.1387)        | 0.2112 (0.12703)        | **0.0150 (0.0055)**    |
> | 60   | 0.1607 (0.1922)       | 0.3929 (0.0202)        | 0.4319 (0.0238)        | 0.3757 (0.15479)        | **0.0172 (0.0046)**    |
> | 70   | 0.3670 (0.0267)       | 0.4566 (0.0446)        | 0.4726 (0.0208)        | 0.35702 (0.02135)       | **0.0272 (0.0103)**    |
>
> ---
>
> ### **Table: 2D Helmholtz Mean Absolute Error(MAE) with Standard Deviation in parantheses.**
>
> | Task | PINN (Dynamic)      | Curr-Reg            | R3                 | Meta-PDE            | LeIn-PINN            |
> |------|----------------------|----------------------|---------------------|----------------------|-----------------------|
> | a₄₄ | 0.0047 (0.0050)      | 0.4131 (0.0264)      | 0.3864 (0.0360)     | 0.2473 (0.1541)      | **0.0013 (0.0004)**   |
> | a₄₅ | 0.2154 (0.3049)      | 0.6172 (0.2846)      | 1.5196 (2.4908)     | 0.4736 (0.3218)      | **0.0029 (0.0006)**   |
> | a₄₆ | 0.0850 (0.1744)      | 0.4050 (0.0056)      | 0.4272 (0.0356)     | 0.4954 (0.3495)      | **0.0050 (0.0009)**   |
> | a₅₅ | 0.1962 (0.2643)      | 0.4106 (0.0105)      | 0.4173 (0.0215)     | 0.5982 (0.3420)      | **0.0058 (0.0035)**   |
> | a₅₆ | 0.0879 (0.1728)      | 0.4390 (0.0683)      | 0.4139 (0.0143)     | 0.4077 (0.0092)      | **0.0081 (0.0015)**   |
> | a₆₆ | 0.1687 (0.2153)      | 0.4346 (0.0685)      | 0.4204 (0.0197)     | 0.4329 (0.0578)      | **0.0150 (0.0107)**   |
>
>
> ---
>
> ### **Table: 2D Navier–Stokes Mean Absolute Error(MAE) with Standard Deviation in parantheses.**
>
> | Re            | PINN (Dynamic)        | Curr-Reg.            | R3                    | Meta-PDE              | LeIn-PINN                |
> |---------------|------------------------|-----------------------|------------------------|------------------------|---------------------------|
> | 600           | 0.2516 (0.4260)        | 0.2133 (0.1783)       | 0.2456 (0.3938)        | 0.3172 (0.0400)        | **0.0057 (0.0005)**       |
> | 800           | 0.0073 (0.0028)        | 0.2150 (0.1753)       | 0.0137 (0.0066)        | 0.2997 (0.0262)        | **0.0058 (0.0009)**       |
> | 1000          | 0.0059 (0.0005)        | 0.2147 (0.1757)       | 0.0426 (0.0428)        | 0.2812 (0.0342)        | **0.0057 (0.0001)**       |
>
>
>
> ---
>
> ### [Visualizations to Justify Quantitative Improvements]
>
> Qualitative results of failures of regular PINNs relative to LeIn-PINNs in challenging PDE domains for **1D Convection**, **2D Helmholtz**, and **2D Navier-Stokes** have been included in Fig. 1a, Fig. 1b, and Fig. 2 in the main paper.
>
> - Fig. 1a, Fig. 1b (center): catastrophic failures of PINN (Dynamic)
> - Fig. 1a, Fig. 1b (right): successful reconstructions by LeIn-PINN
> - Fig. 2e: PINN (Dynamic) prediction of pressure field for Navier–Stokes
> - Fig. 2f: LeIn-PINN prediction (far more faithful to ground truth)
>
> These qualitative plots strongly reinforce our claim that LeIn-PINNs overcome the catastrophic failures that PINNs face in challenging PDE domains.
>
> ### References
>
> 1. Qin, Tian, et al. "Meta-pde: Learning to solve pdes quickly without a mesh." arXiv preprint arXiv:2211.01604 (2022).

---

### Official Review · Reviewer_qwKR · 2025-11-01

**Soundness:** 3
**Presentation:** 3
**Contribution:** 3
**Rating:** 6
**Confidence:** 4

**Summary:**

This work introduces a novel Physics-informed Neural Network (PINN) training methodology called Learned Initialization (LeIn), specifically designed to mitigate frequent catastrophic convergence failures in challenging Partial Differential Equation (PDE) domains. The paper rigorously explores the critical effect of learned weight initialization on PINN training dynamics, and they introduced Invariance Encoding and Gated Layer-wise Optimization to address convergence failures across diverse PINN-based PDE problems. The authors conducted both quantitative and qualitative experiments, to address three core questions: the role of learned initialization in mitigating catastrophic training failures, the performance comparison against state-of-the-art (SoTA) approaches, and the individual effect of invariance encoding and gated layer-wise optimization.

**Strengths:**

The proposed method is simple but powerful, effectively overcoming the common convergence failures of PINNs when applied to hard or challenging PDE equations. By distilling invariant physics into the initial weights, LeIn provides a superior starting point that stabilizes the optimization process.

The rigorous experiments across 1D and 2D PDE domains, demonstrate that LeIn-PINNs significantly outperform state-of-the-art approaches.

The work includes ablation studies to isolate and confirm the advantage of each proposed component, namely Invariance Encoding and Gated Layer-wise Optimization. Furthermore, the paper provides a detailed Hessian Eigenvector Landscape analysis and evidence of reduced spectral bias to support the strength of the proposed method.

**Weaknesses:**

1. Missing PDE Background in Introduction: The Introduction and abstract introduce the problems (Convection, Helmholtz, Navier-Stokes) without providing explanation. Concepts like transport phenomena (1D convection) and wave propagation (2D Helmholtz) are only briefly mentioned in the Results & Discussion section. To enhance reader comprehension, it is better to include the Introduction a concise, clear explanation of these PDEs.

2. Incomplete Related Work: While the paper references Wang et al., 2022a, the comparative analysis in the Method section relies on the methodology presented in Wang et al., 2021a. Introducing and briefly explaining the 2021a in the Related Work section will allow readers to understand the comparison baseline better.

3. Ambiguous Terminology and Definition: The consistent use of subjective terms like "stiff," "challenging," and "hard" to describe PDE configurations throughout the paper lacks rigorous definition. Providing a clear, quantifiable criterion or metric for distinguishing between "easy" and "hard" PDE problems is necessary.

4. Equation and Abbreviation Clarification:
- The full name for the abbreviation MAML (Model-Agnostic Meta-Learning) should be provided upon its first mention in the Related Work section.
- In the Problem Formulation section, the explanation of the lower letter b within Equation (1) is required for complete mathematical clarity.

5. Lack of Specificity: In the Problem Formulation section, the Invariance Encoding component relies on a "set of relatively easier configurations". The criteria and mechanism for generating or selecting this set are not elaborated.

6. Insufficiency in Claiming Novelty: If the Gated Layer-wise Optimization is a novel contribution of this paper, the authors must assert its originality more forcefully in the Introduction and Method sections, explicitly stating that it is the first application of its kind used to address PINN convergence failures.

7. Clarity on Initialization Baseline:
- The paper claims PINN address poor convergence associated with conventional initialization schemes (Xavier/Kaiming). Providing how poor the convergence with Xavier, Kaiming or other initialization schemes will support the strength of LeIn-PINN.
- The authors should also clarify the paper's contribution regarding weight initialization: Is LeIn-PINN the first work to explore initialization of weight in the PINN training?

8. Deeper Explanation of Landscape Analysis: In Section 4.1 (Hessian Eigenvector Landscape analysis), the claim that LeIn-PINNs achieve a "smoother landscape that supports convergence to better minima" is too succinct. Providing more explanation like, the presence of a "rugged surface with deep valleys" often leads to undesirable local minima, will help convincing readers understand better.

**Questions:**

See Weaknesses.

---

> ### Author Response · Authors · 2025-11-25
> **W1-W6. Response to reviewer qwKR**
>
> We thank the reviewer for their time and their insight. Please find our responses below.
>
> ## `W1. Missing PDE Background in Introduction`
>
> Thank you for the suggestion. To clarify the PDE domains considered, we have infact added clear descriptions in a separate (self-contained) section titled “Dataset Description and Experimental Setup”. We have confined mentions of specific PDE domains to this section to avoid confusion.
>
>
> ## `W2. Incomplete Related Work`
>
> Thank you, we have added the brief synopsis of the mentioned paper in the related work.
>
>
> ## `W3. Ambiguous Terminology and Definition`
>
> We have included a “Dataset Description and Experimental Setup” section where we have explicitly demarcated easy vs. hard settings considered as well as provided references where more information about properties of specific domains can be obtained. Finally, we have also provided more background about each of the PDE domains considered along with the accompanying challenges that PINNs face in modeling PDE dynamics.
>
>
> ## `W4. Equation and Abbreviation Clarification`
>
> Thank you, we have addressed these points.
>
>
> ## `W5. Lack of Specificity`
>
> We have evaluated our proposed method on three variegated PDE domains (1D Convection, 2D Helmholtz and 2D Navier Stokes) and have relied on previous work [1,2,3] to define regimes of easy task ranges for training and hard tasks for evaluation. We have added a new section (Section 4) titled `Dataset Description and Experimental Setup` detailing the easy and hard task regimes for each PDE domain along with references.
>
> ### References
>
> 1. Krishnapriyan, Aditi, et al. "Characterizing possible failure modes in physics-informed neural networks." Advances in Neural Information Processing Systems 34 (2021): 26548–26560.
>
> 2. Wang, Sifan, Yujun Teng, and Paris Perdikaris. "Understanding and mitigating gradient flow pathologies in physics-informed neural networks." SIAM Journal on Scientific Computing 43.5 (2021): A3055–A3081.
>
> 3. Jongmok Lee, Seungmin Shin, Taewan Kim, Bumsoo Park, Ho Choi, Anna Lee, Minseok Choi, and Seungchul Lee. "Physics-informed neural networks for fluid flow analysis with repetitive parameter initialization." Scientific Reports 15(1):16740, 2025.
>
>
> ## `W6. Insufficiency in Claiming Novelty`
>
> Thank you for the suggestion. The gated layer-wise optimization employed in the learned initialization mechanism is indeed a novel contribution, and we have updated text in the introduction to more explicitly communicate our contributions.

---

> ### Author Response · Authors · 2025-11-25
> **W7, W8. Response to reviewer qwKR.**
>
> ## `W7. Clarity on Initialization Baseline`
>
> Thank you. We agree that results indicating poor convergence with Xavier and Kaiming will indeed serve to strengthen our narrative. Our results in fact already demonstrated this point in three different experiments, and we have added an experiment in response to your thoughtful suggestion. We summarize all four experiments and results below.
>
>
> ### (i) Results in Table. 1–3 show empirical superiority of Learned Initialization
>
> Specifically, all results in the main paper have been conducted with PINN variants that employ Xavier initialization (in line with the popular best practice) and results in Table 1–3 already demonstrate quantitative empirical superiority of LeIn-PINNs over multiple state-of-the-art PINN variants which all employ Xavier initialization.
>
> In line with your suggestion, we have also added PINN(Dynamic) variant with Kaiming initialization (results in table below) where we show that PINN(Dynamic) with LeIn-Initialization outperforms variants with Xavier and Kaiming initialization (results have also been added to Appendix E.5, Table 11.)
>
> ### PINN (Dynamic): Xavier vs Kaiming vs LeIn
>
> | Task | PINN (Dynamic) Xavier | PINN (Dynamic) Kaiming | PINN (Dynamic) LeIn |
> |------|-----------------------|--------------------------|-----------------|
> | 30   | **0.0071 (0.0023)**   | 0.0092 (0.00256)         | 0.0092 (0.0025) |
> | 40   | 0.0126 (0.0051)       | 0.01582 (0.00288)        | **0.0122 (0.0045)** |
> | 50   | 0.0226 (0.0072)       | 0.01762 (0.00232)        | **0.0150 (0.0055)** |
> | 60   | 0.1607 (0.1922)       | 0.02098 (0.00572)        | **0.0172 (0.0046)** |
> | 70   | 0.3670 (0.0267)       | 0.07076 (0.06444)        | **0.0272 (0.0103)** |
>
>
>
> ### (ii) Loss-Landscape Results Show Smooth Hessian
>
> Our loss-landscape analysis depicted in Fig. 4 clearly demonstrates that Learned-Initialization has smoother loss landscapes compared to random initialization thereby proving more amenable for gradient-based optimization.
>
>
> ### (iii) Qualitative Results
>
> We demonstrate plots for challenging domains for 1D Convection, 2D Helmholtz as well as 2D Navier-Stokes, showing the full predicted domains yielded by randomly initialized PINN in Fig. 1a (center), Fig. 1b (center) and Fig. 2e respectively, vs. predictions yielded by LeIn-PINNs in Fig. 1a (right), Fig. 1b (right) and Fig. 2f respectively.
>
>
> ### (iv) Effect of Learned Initialization on Other State-of-the-Art Models
>
> To further corroborate our point, we also test the effect Learned Initialization has on the R3 model, a state-of-the-art PINN variant proposed solely to alleviate catastrophic failures in PINNs.
>
> Specifically, in Appendix E.6, Table 14, we compare performance of R3 with Xavier Initialization and a variant with Learned Initialization and evaluate on all extrapolation tasks in 1D convection. We can clearly see that LeIn-R3 variant outperforms the R3 variant with Xavier initialization for ALL evaluated extrapolation settings.
>
> ### R3: Xavier Init vs Learned Init
>
> | Task | R3 (Xavier Init.) | R3 (Learned Init.) | MAE Reduction (%) |
> |------|--------------------|--------------------|--------------------|
> | 30   | 0.0067             | **0.0055**        | 17.91%            |
> | 40   | 0.0678             | **0.0106**        | 84.37%            |
> | 50   | 0.3224             | **0.1050**        | 67.43%            |
> | 60   | 0.4319             | **0.3670**        | 15.02%            |
> | 70   | 0.4726             | **0.3384**        | 28.39%            |
>
>
> ### Clarification on Contribution
>
> Yes, LeIn-PINN is the first and (to the best of our knowledge) ONLY work to explore learned initialization strategies to alleviate catastrophic training failures in PINN models. In response to your suggestion above, we have explicitly indicated (by using exactly this phrase) this as one of our contributions.
>
>
> ## `W8. Deeper Explanation of Landscape Analysis`
>
> Thank you, we have added additional text (and an accompanying citation) in line with your suggestion to further clarify the results in the loss landscape analysis.

---

### Official Review · Reviewer_PMRR · 2025-11-01

**Soundness:** 2
**Presentation:** 2
**Contribution:** 2
**Rating:** 2
**Confidence:** 4

**Summary:**

The paper proposes LeIn, a meta-learned initialization for PINNs aimed at reducing convergence failures on challenging PDEs (e.g., stiff regimes or difficult inverse problems). LeIn has two components:
(1) Invariance Encoding (IE): meta-updates over a distribution of easy PDE instances to encode common physics into the weights, (2) Gated Layer-wise Optimization (GLO) — a layer-gating schedule during meta-updates that gradually unlocks deeper layers to stabilize optimization. After meta-training, the learned weights serve as initialization for standard PINN training on a hard target instance.

**Strengths:**

1. Once the initialization is learned, downstream training uses the standard PINN objective with minimal code changes.

2. IE targets cross-instance invariants; GLO acknowledges heterogeneous loss dynamics across layers and mitigates instability early in training.

3. The method shows consistent gains on benchmark PDEs where vanilla PINNs are known to fail.

**Weaknesses:**

1. The baseline suite appears limited; several baselines seem to time out. A budget-controlled comparison with stronger baselines (curriculum, transfer initialization, and meta-learning methods such as MAML/Reptile and Hyper-LR-PINN) is needed for fairness.

2. Meta-training cost & data design: The method depends on a well-curated distribution of “easy” tasks and incurs one-time meta-training overhead. The paper would benefit from a sensitivity study to the coverage/quality of this task distribution.

3. Ablations of GLO: The paper motivates hard gating, but does not fully explore whether soft gating, layer-wise learning-rate schedules, or alternative unlock schedules (cosine, exponential) would perform as well or better.

**Questions:**

1. Since the core idea is meta-initialization, please compare against generic meta-learning methods (e.g., MAML, Reptile) and meta-PINN approaches (e.g., Hyper-LR-PINN) under the same task distribution and collocation. Report failure rate across seeds, median/mean errors, and time-to-target-accuracy.

2. Test higher-dimensional PDEs (e.g., 3D Navier–Stokes) and complex boundary/geometry to substantiate generality. Even small-scale 3D or simplified geometries would strengthen the claim.

3. Since each unseen instance still needs fine-tuning, please report (i) adaptation steps/time, (ii) final storage (base LeIn weights + per-instance deltas). Consider an adapter/LoRA variant to quantify memory savings.

4. Operator-learning baselines (e.g., FNO/DeepONet) or conditional PINNs for parameterized families—while not apples-to-apples (data-driven vs physics-informed), they are practical alternatives for multi-instance generalization. A brief discussion with controlled experiments would strengthen positioning.

5. Please standardize notation for parameters/coefficient vectors and clearly define the “easy” task distribution in the main text (not only in appendix).

---

> ### Author Response · Authors · 2025-11-25
> **Q1. Response to reviewer PMRR.**
>
> We thank the reviewer for their time and the insightful questions. Please find our responses below.
>
> ## `Q1 Addressing Comparisons with Generic Meta-Learning and Meta-PINN Baselines`
>
> **[Comparison with Generic Meta-Learning]:** Thank you for your suggestion, we had indeed compared with generic meta-learning methods like MAML in our original paper submission.  Specifically, our paper covered an ablation study as part of RQ3 (Section 5.3 - Fig. 5).  In Fig. 5, the variant without the gated layer optimization (i.e., “LeIn-PINN w/o GLO”) corresponds to a generic MAML model. We apologize for this confusion in the paper and have added text to the first paragraph of Section 5.3, clarifying that “LeIn-PINN w/o GLO” is a generic MAML model.
> In addition, we have now also included another baseline Meta-PDE [1] as requested by reviewer 26a7. Meta-PDE was evaluated under the same training conditions across all three PDE systems and has been added to Tables 1–4 in the main paper. Consistently, LeIn-PINNs outperforms Meta-PDE by **86.7%** according to MAE metric, in the more challenging extrapolation regimes we investigate.This aligns with our hypothesis that controlling layer-wise optimization dynamics is essential for stable PINN training. This control is provided by our proposed GLO mechanism, but is absent in Meta-PDE.
>
> **[Performance Across Random Seeds + Median Errors]:** In our original paper, the box plots shown (Fig. 5) as part of the ablation analysis were obtained by executing the results (on all 3 PDE domains) on 5 different random seeds and reporting the median, standard deviation statistics. Our results show that LeIn-PINN, when equipped with both GLO and invariance encoding (IE), delivers stronger performance and stability than just employing the generic MAML (i.e., only IE but no GLO) for learning the initialization. This is especially apparent, if we look at the most challenging settings (e.g., Fig. 5c, 2D Navier-Stokes Re = {800, 1000}) where we clearly see that generic MAML model (i.e., “LeIn-PINN w/o GLO”) yields unstable performance reporting high standard deviations while also yielding worse median performance than LeIn-PINNs.
>
> **[Convergence of Generic MAML vs. LeIn-PINN]:** All variants (i.e., “LeIn-PINN”, “LeIn-PINN w/o GLO” and “LeIn-PINN w/o (IE, GLO)”) were trained under the same training and computational budgets (i.e., 6K steps for invariance encoding and 50K steps for fine-tuning a LeIn-PINN on unseen PDE domain), with matched gradient-update settings.
> So all our comparisons reflect genuine performance differences rather than unequal resource allocation. Overall as can be observed from the final performances of PINNs on fine-tuned (extrapolation and interpolation) domains, LeIn-PINNs converge to far superior solutions compared to generic MAML trained models.
>
> **[Additional Results: Mean/Std Error Reporting].** Finally, we have also added an appendix section E4. titled “Extended Results: Mean and Standard Deviation” (see Table 8-10) to capture performance of all models (i.e., LeIn-PINN as well as baseline models) for all 3 PDE domains. In each table, the mean and standard deviation performance of each model across all 5 random seeds are reported using a mean-absolute error (MAE) metric. We have also included one of these tables (2D Helmholtz) below for ease of analysis. Across all tables, it is apparent that LeIn-PINN models have not only the best mean performance across random seeds but also are the most stable (i.e., least standard deviation in performance) across a majority of the investigated cases.
>
> ### Mean Absolute Error (MAE) on the 2D Helmholtz PDE across varying aₓᵧ in the extrapolation setting
> Values report the mean MAE across 5 random seeds, with corresponding standard deviation reported in parentheses.
>
>
> | Task | PINN (Dynamic)      | Curr-Reg            | R3                 | Meta-PDE            | LeIn-PINN            |
> |------|----------------------|----------------------|---------------------|----------------------|-----------------------|
> | a₄₄ | 0.0047 (0.0050)      | 0.4131 (0.0264)      | 0.3864 (0.0360)     | 0.2473 (0.1541)      | **0.0013 (0.0004)**   |
> | a₄₅ | 0.2154 (0.3049)      | 0.6172 (0.2846)      | 1.5196 (2.4908)     | 0.4736 (0.3218)      | **0.0029 (0.0006)**   |
> | a₄₆ | 0.0850 (0.1744)      | 0.4050 (0.0056)      | 0.4272 (0.0356)     | 0.4954 (0.3495)      | **0.0050 (0.0009)**   |
> | a₅₅ | 0.1962 (0.2643)      | 0.4106 (0.0105)      | 0.4173 (0.0215)     | 0.5982 (0.3420)      | **0.0058 (0.0035)**   |
> | a₅₆ | 0.0879 (0.1728)      | 0.4390 (0.0683)      | 0.4139 (0.0143)     | 0.4077 (0.0092)      | **0.0081 (0.0015)**   |
> | a₆₆ | 0.1687 (0.2153)      | 0.4346 (0.0685)      | 0.4204 (0.0197)     | 0.4329 (0.0578)      | **0.0150 (0.0107)**   |
>
> ### References
>
> 1. Qin, Tian, et al. "Meta-pde: Learning to solve pdes quickly without a mesh." arXiv preprint arXiv:2211.01604 (2022).

---

> ### Author Response · Authors · 2025-11-25
> **Q2. Response to reviewer PMRR.**
>
> ## `Q2 On Generality Beyond 2D: Higher-Dimensional PDEs and Complex Geometries`
>
> Thank you for the suggestion. Our main claim in the paper is that our proposed Learned-Initialization Physics-Informed Neural Network (LeIn-PINN) alleviates catastrophic failures experienced by PINNs in challenging PDE domains. Our choice of the three PDE domains is based on previous papers [1, 2, 3] that focus on the same problem of alleviating PINN failures in challenging PDE domains. We believe that our current experimental evaluation on three diverse PDE domains itself is rigorous and strong enough to demonstrate our main claim. Specifically, our tests on 1D Convection (that models transport phenomena), 2D-Helmholtz (modeling wave propagation) and 2D-Navier-Stokes (involving coupled convective and diffusive dynamics) already demonstrate (in extremely challenging conditions) the need for learned initialization to not only alleviate PINN catastrophic failures but also for improved extrapolation performance.
>
> ### References
>
> 1. Wang, Sifan, Yujun Teng, and Paris Perdikaris. "Understanding and mitigating gradient flow pathologies in physics-informed neural networks." SIAM Journal on Scientific
>    Computing 43.5 (2021): A3055-A3081.
>
> 2. Krishnapriyan, Aditi, et al. "Characterizing possible failure modes in physics-informed neural networks." Advances in neural information processing systems 34 (2021): 26548–26560.
>
> 3. Daw, Arka, et al. "Mitigating Propagation Failures in Physics-informed Neural Networks using Retain-Resample-Release (R3) Sampling." Proceedings of Machine Learning Research 202 (2023): 7264–7302.

---

> ### Author Response · Authors · 2025-11-25
> **Q3-Q5. Response to reviewer PMRR.**
>
> ## `Q3 Adaptation Cost Analysis: Fine-Tuning Steps, Runtime, and Storage Footprint`
>
> We thank the reviewer for raising this important point. The architecture employed by LeIn-PINNs is EXACTLY the same as that of vanilla PINNs. The same architecture is first subjected to the “Invariance Encoding” stage followed by a PDE-specific fine-tuning stage. The need for PDE-specific (i.e., per-instance) fine-tuning is consistent with the original vanilla PINN formulation which requires fine-tuning for each new PDE domain.
>
> ### i. Adaptation Steps / Time
>
> Our LeIn-PINN pipelines have been executed for a total of 56,000 training steps with 6000 steps for invariance-encoding (IE) + GLO and 50,000 steps for fine-tuning.
>
> To test the effect of number of adaptation steps, we execute an experiment on 1D convection where all baseline models have also been executed for 56K steps. We find that this increased number of updates / increased time for adaptation has little effect in changing the results and performance is still strongly in favor of LeIn-PINN models indicating that LeIn-PINNs converge higher quality solutions compared to baselines, given the same convergence time / same number of adaptation steps.
>
> ### 1D Convection (56K Steps)
>
> | Task | Curr Reg (56K) | R3 (56K) | LeIn-PINN  |
> |------|----------------|----------|------------|
> | 50   | 0.3004         | 0.3291   | **0.0150** |
> | 60   | 0.4015         | 0.4356   | **0.0172** |
> | 70   | 0.4208         | 0.4565   | **0.0272** |
> | 80   | 0.4738         | 0.4648   | **0.2490** |
>
> ### ii. Memory and Storage Footprint during Fine-Tuning
>
> Owing to LeIn-PINNs employing the same architecture as vanilla PINNs, LeIn-PINNs during fine-tuning have EXACTLY the same memory and storage footprint as vanilla PINNs and hence efficient low-rank adaptation techniques like LoRA are not necessary.
>
> To reinforce our point, we have included (in the table below) the exact memory and training time of PINN and LeIn-PINN fine-tuning for 1D convection. From the table, we see that time and memory footprints for LeIn-PINN (Fine-Tuning) and “PINN Training” of a randomly initialized PINN are identical.
>
> Additionally, the only added cost of LeIn-PINN is the one-time pre-training (i.e., invariance encoding + GLO) cost. Once the initialization is learned, it can be reused to fine-tune across multiple PDE tasks, thereby amortizing this overhead.
>
> ### Memory and Time Comparison
>
> | Stage / Method                     | Time (min) | Peak GPU Memory (MB) |
> |-----------------------------------|------------|------------------------|
> | LeIn-PINN (Invariance Encoding+GLO) | 3.1        | 740                    |
> | LeIn-PINN (Fine-Tuning)           | 10.4       | 540                    |
> | Xavier Init. PINN Training        | 10.4       | 540                    |
>
>
> ---
>
>
> ## `Q4 Comparison to Conditional PINN for Parameterized PDE Families`
>
> Thank you for your suggestion. To address comparisons with conditional PINN for parameterized PDE families, we have also added results comparing with a recent conditional PINN paper, specifically, P²INN [1] that takes into account PDE parameters as an input condition, and addresses the task of modeling parameterized PDE families.
>
>
> Specifically, we have conducted experiments on the 1D convection dataset in the extrapolation setting, in each case evaluating across 5 random seeds. The mean absolute error (MAE) results (table below) report the average across these 5 random seeds along with standard deviation reported in parentheses. From the table, it is clear that the LeIn-PINN model significantly out-performs P²INN by **93.6%**, across all extrapolation settings.
>
>
> This comparison has also been included in the appendix E.5 (Table 12) of the revised manuscript and provided below for ease of analysis. We see parameterized PINN fail to extrapolate to difficult settings compared to the LeIn-PINN model.
>
>
> ### 1D Convection: P²INN vs LeIn-PINN
>
> | Task | P$^2$INN        | LeIn-PINN           |
> |------|-----------------|---------------------|
> | 30   | 0.2259 (0.0202) | **0.0092 (0.0025)** |
> | 40   | 0.2485 (0.0091) | **0.0122 (0.0045)** |
> | 50   | 0.2561 (0.0023) | **0.0150 (0.0055)** |
> | 60   | 0.2559 (0.0014) | **0.0172 (0.0046)** |
> | 70   | 0.2579 (0.0009) | **0.0272 (0.0103)** |
>
>
> ### References
>
> 1. Park, Noseong, et al. "Parameterized physics-informed neural networks for parameterized PDEs." 41st International Conference on Machine Learning, ICML 2024.
>
> ---
>
> ## `Q5 Clarifications on Notation and Definition of the Easy-Task Distribution`
>
> Thank you for the suggestion. We have added Section 4 titled, “Dataset Description and Experimental Setup” in the updated main paper to describe the PDE domains and the “easy” task distributions in each case. Further, we have worked to standardize notations in line with your suggestion to achieve consistency across problem formulation and PDE descriptions.

---

### Author Response · Authors · 2025-12-03
**Overall Summary & List of Changes During Rebuttal**

The main claim of our paper is that `intelligent initialization of network weights prior to training, helps physics-informed neural networks (PINN) overcome catastrophic failures, while modeling challenging partial differential equation (PDE) domains.` To this end, we propose Physics Informed Neural Networks with Learned-Initialization, that we abbreviate as **LeIn-PINN.**

**Note 1:** Ours is the first (and to the best of our knowledge) the only work to have investigated the ability of learned initialization to alleviate the catastrophic failures that PINNs commonly succumb to while modeling challenging PDE domains.

**Note 2:** We have also introduced a new “Gated Layer-wise Optimization” (GLO) method to overcome challenging training dynamics inherent in PINNs, and have demonstrated the positive effect of the GLO method in improving performance, through a rigorous ablation analysis (Section 5.3, Fig. 5).

**Note 3:** Previous papers [1], that have investigated the ability of PINNs to overcome catastrophic failures, have done so on simpler configurations of the 1D Convection, 2D Helmholtz domains (which are two common domains of comparison). Our current work pushes this investigation to far more complex configurations of the same PDE domains (in addition to modeling with the challenging 2D Navier-Stokes domain) and demonstrates clear failures of numerous state-of-the-art baselines that were proposed explicitly with the goal of overcoming PINN catastrophic failures. In contrast, our experiments demonstrate that LeIn-PINNs succeed and achieve the best performance without succumbing to catastrophic failures in all evaluated PDE domains.

**Note 4:** All our experiments have been evaluated across 5 random seeds using mean-absolute-error (MAE) to evaluate prediction quality. The mean MAE and standard deviations across the 5 runs for each model on each PDE domain, have been reported in the appendix (specifically, Appendix E.4, Tables 8,9,10) with only the mean MAEs being reported in the main paper (Tables 1, 2, 3) owing to space constraints. Overall our investigations have revealed that our proposed *LeIn-PINN models achieve a mean performance improvement of **89%** across all state-of-the-art baselines.*

---

**Differentiation from Previous Work:** Reviewer `26a7` has stated that: “meta-learning for PINNs has been studied in papers [2,3,4,9] in the context of PINNs. It would be good to more explicitly discuss the shortcomings of these existing works with respect to improved PINN training.” The reviewer has further requested us to employ one of the stated methods as a baseline to strengthen our claim.

We would first like to state that meta-learning is NOT the point of our paper and the main point of our paper is to demonstrate that intelligent initialization of PINN weights, can be used to overcome catastrophic failures of PINNs in challenging PDE domains. Ours is a crucial contribution because previous works have employed complementary strategies like different deep learning architectures [1], different sampling methodologies [1,5] and curriculum learning [6] based training strategies to overcome catastrophic failures but `NONE of the previous works have investigated the effect of intelligent initialization of the PINN model weights, on its ability to overcome catastrophic failures in challenging PDE domains.` Further, `no work, to the best of our knowledge has investigated as challenging a set of PDE configurations in each domain as we have undertaken in the current paper`. Even in such a rigorous and challenging set of PDE configurations, our proposed LeIn-PINNs have outperformed ALL compared baselines [1,2,5,6] across ALL 3 PDE domains.
To explicitly address reviewer `26a7`’s request, we have also employed Meta-PDE \[2\] (one of the papers recommended by the reviewer) as a baseline (details in `26a7`’s rebuttal section).

---

> ### Author Response · Authors · 2025-12-03
> **List of Changes During Rebuttal**
>
> # A. Experimental Changes
>
> ## `Response to Reviewer` **PMRR**
>
> **Mean and variance reporting for all experiments**
> *(Appendix E.4, Tables 8–10)*
> The reviewer requested that mean and variance metrics be reported, to assess stability across random seeds. We added these statistics (computed over 5 random seeds) for all baselines and for LeIn-PINN. The results show that LeIn-PINN performance remains consistent across different seeds.
>
> **Comparison with conditional PINNs for parameterized families (P²INN [8])**
> *(Appendix E.5, Table 12)*
> The reviewer requested a comparison with conditional PINNs for parameterized PDE families. We added the P²INN baseline and showed that LeIn-PINN achieves lower MAE under equal training conditions. P²INN struggles in the harder extrapolation regimes included in our setup.
>
> **Sensitivity of the learned-initialization step to task distribution**
> *(Appendix E.7, Tables 15–16)*
> In response to the reviewer's inquiry regarding how the performance of the learned-initialization step, depends on the difficulty of tasks chosen, we added an ablation analysis. Specifically, we conducted this ablation by varying the distribution of easy tasks on which the learned-initialization step was trained. After fine-tuning on challenging PDE domains, we found that LeIn-PINN remained the best-performing model regardless of how the initialization tasks were sampled. Explanatory text has been included in Appendix E.7.
>
> ---
>
> ## `Response to Reviewer` **qwKR**
>
> **Comparison with standard initializations (Xavier, Kaiming)**
> *(Appendix E.5, Table 11)*
> The reviewer requested a comparison between learned initialization and standard schemes. We ran an ablation on the 1D Convection domain and found that learned initialization consistently yields lower error than Xavier and Kaiming. We have reported results and added corresponding explanatory text in Appendix E.5 .
>
> ---
>
> ## `Response to Reviewer` **26a7**
>
> **Baseline comparison with a meta-learning method**
> *(Tables 1–3)*
> As requested, we evaluated the Meta-PDE [2] baseline. LeIn-PINN outperforms Meta-PDE across all three PDE domains, with an average improvement of **86.7%**.
>
> **Inspection of layer-wise initialization before task-specific training**
> *(Appendix E.5, Table 13)*
> The reviewer requested for an examination of the learned initialization weights before any task-specific fine-tuning was performed. To address this request, we compared the layer-wise parameter distributions of a randomly initialized PINN and a PINN initialized through the learned initialization step. We observed strong dissimilarity in shallower layers and increasing similarity in deeper layers. The strong dissimilarity in shallower layers was primarily attributable to higher variance of weight distribution in the learned-initialization weights (also reported in Table 13). The higher variance in shallower layers aligns with prior findings [7] that larger variance in weight distributions increases neural network expressivity. This suggests that LeIn-PINNs benefit from greater expressive capacity (as a result of learned initialization). This is reinforced by the performance gains reported in our experimental comparisons (Tables 1, 2, 3).
>
> ---
>
> # B. Textual Changes
>
> ## `Response to Reviewer` **PMRR**
> - Updated text in Appendix (E.7, Line 1348-1364) to better reference and highlight sensitivity analysis of LeIn-PINN with respect to different range of easy tasks.
>
> ## `Response to Reviewer` **qwKR**
> - Further clarified the novelty of our contribution in the introduction (Section 1, Line 57-64).
> - Improved clarity in the problem formulation and added missing notation definitions (Section 3, Line 102-121).
> - Added a *Dataset Description* section and defined the easy and hard PDE configurations (Section 4, Line 195-232).
> - Improved the explanation to better support our claim regarding the loss landscape of LeIn-PINNs being superior to that of randomly initialized PINNs and grounded our explanation, in insights from previous research which were also included as references. (Section 5, Line 385-386).
> - Added text highlighting the outcome of extended-training of baselines in comparison with LeIn-PINNs and how LeIn-PINNs still outperformed all baselines despite baselines having been subjected to additional training. (Section 5, Line 421-424).
>
> ## `Response to Reviewer` **SMmg**
> - Added explanatory text in the appendix describing our comparison with parameterized PINNs, along with extended-training results to ensure fairness in baseline evaluation (Appendix E.5, Line 1269-1271).
>
> ## `Response to Reviewer` **26a7**
> - Added clarification in the main text (Section 2, Line 76; Line 92-98) explaining how our method compares to meta-learning approaches and differentiating our method from previous approaches.

---

> ### Author Response · Authors · 2025-12-03
> **References**
>
> 1. Wang, Sifan, Yujun Teng, and Paris Perdikaris. "Understanding and mitigating gradient flow pathologies in physics-informed neural networks." SIAM Journal on Scientific Computing 43.5 (2021): A3055-A3081.
>
> 2. Qin, Tian, et al. "Meta-pde: Learning to solve pdes quickly without a mesh." arXiv preprint arXiv:2211.01604 (2022).
>
> 3. Psaros, Apostolos F., Kenji Kawaguchi, and George Em Karniadakis. "Meta-learning PINN loss functions." Journal of computational physics 458 (2022): 111121.
>
> 4. Toloubidokhti, Maryam, et al. "Dats: Difficulty-aware task sampler for meta-learning physics-informed neural networks." The Twelfth International Conference on Learning Representations. 2023
>
> 5. Daw, Arka, et al. "Mitigating propagation failures in physics-informed neural networks using retain-resample-release (r3) sampling." Proceedings of the 40th International Conference on Machine Learning. 2023.
>
> 6. Krishnapriyan, Aditi, et al. "Characterizing possible failure modes in physics-informed neural networks." Advances in neural information processing systems 34 (2021): 26548-26560.
>
> 7. Yang, Ge, and Samuel Schoenholz. "Mean field residual networks: On the edge of chaos." Advances in neural information processing systems 30 (2017).
>
> 8. Park, Noseong, et al. "Parameterized physics-informed neural networks for parameterized PDEs." 41st International Conference on Machine Learning, ICML 2024. ICML, 2024.
>
> 9. Hemachandra, Apivich, et al. "PIED: Physics-Informed Experimental Design for Inverse Problems." arXiv preprint arXiv:2503.07070 (2025).

---

### Meta-Review · Area_Chair_xaGa · 2026-01-05

**Summary:**

The paper received overall negative evaluations, with several major concerns remaining unresolved after the rebuttal. While some clarity and presentation issues were addressed, key technical requests such as budget-controlled comparisons, stronger and more diverse baselines, ablations and justification of the GLO component, and a concrete definition of "easy" versus "hard" tasks were not adequately addressed. The reviewers’ expected final scores are 2, 6, 4, and 4, yielding an average score of 4.0 with moderate to high reviewer confidence. Given the persistence of these core concerns and the lack of score movement, I recommend rejection.

**Reviewer Concerns:**

Reviewer PMRR

- Concerns that are potentially addressed:

  - Adaptation cost analysis: The authors’ response could partially address this concern, although the analysis remains limited.

  - Comparison with NO family or conditioned PINNs: The authors added new experiments with P2INNs, which could partially address this point while P2INNs are not neural operators.

- Concerns that are likely not addressed:

  - Budget-controlled comparisons with stronger baselines: The authors did not provide the requested budget-controlled comparisons, such as limiting fine-tuning to 50,000 iterations.

  - Sensitivity analysis of task distribution coverage and quality: No response or experimental analysis was provided.

  - Ablation of GLO: This request was not addressed.

  - Comparison with generic meta-learning methods and meta-PINN approaches: The added comparison with meta-PDE, which is MAML-based, effectively corresponds to the proposed method without GLO and does not constitute a substantially stronger baseline.

  - Generality beyond 2D: The authors stated that generalization to 3D problems or complex geometries is not a primary focus, which is unlikely to fully satisfy the reviewer.

  - Definition of "easy" tasks: While specific parameter values were provided, no clear definition or criteria were given.

Reviewer qwKR

- Concerns that are potentially addressed:

  - Missing PDE background and related work

  - Clarification of equations, abbreviations, and terminology

  - Improved specificity in claims

  - Clarification of initialization baselines

  - Deeper explanation of the loss landscape analysis

  - Improved discussion of novelty claims

- Concerns that are likely not addressed:

  - Ambiguous terminology and definitions: The notions of “easy” and “hard” tasks remain subjective, especially given recent advances in PINN architectures and optimization methods.

  - Clarification of contributions: The claim that LeIn-PINN is the first and only work exploring learned initialization to mitigate catastrophic PINN failures remains debatable and insufficiently justified.

Reviewer SMmg

- Concerns that are potentially addressed:

   - Variance and visualization: The authors provided the requested information.

   - Fairness of baseline comparisons: Additional experiments with a revised training protocol and pre-training on easy tasks were provided, although full experimental details remain unclear and could prompt follow-up questions.

- Concerns that are likely not addressed:

     - Motivation for GLO: The response relies primarily on intuition and plausible explanations rather than logical or theoretical justification.

     - GLO hyperparameter tuning and sensitivity: No experimental results were provided to address this concern.

Reviewer 26a7

 - Concerns that are potentially addressed:

     - Error bars: Standard deviations are now reported.

     - Time and resource usage: Additional information was provided, although it remains unclear whether 6,000 epochs correspond to outer-loop iterations in the bilevel optimization. If so, comparisons using 56,000 training epochs for baselines may raise fairness concerns.

      - Visualization of learned initialization: Additional results were provided, but their connection to broader scientific insight remains unclear.

 - Concerns that are likely not addressed:

    - Motivation for learned initialization: The response does not convincingly explain why learned initialization alone should resolve PINN failure modes, especially since other meta-initialized PINNs still fail in the authors’ own experiments.

      - Role of GLO: Although GLO appears to play a critical role, its contribution is not clearly articulated or justified.

      - Stronger baselines: The added meta-PDE baseline is not substantially stronger, as it effectively removes GLO from the proposed method.

      - Comparison with advanced PINN methods: No response was provided.

      - Denser parameter distribution: No response was provided.

       - Concrete definition of "easy" and “hard” tasks: Only specific parameter choices are given, without a principled definition or criteria.

        - Motivation for gating: This concern was not directly addressed, and related responses elsewhere do not provide clear justification.


Common concerns across reviewers:

- Lack of strong and fair baseline comparisons, including missing budget-controlled experiments and comparisons with more advanced PINN and meta-learning methods.

- Insufficient justification and analysis of the GLO and gating mechanisms, with no ablation studies or theoretical grounding to isolate their contributions.

- Unclear and subjective definition of "easy" and "hard" tasks, with only specific parameter choices provided rather than principled criteria.

- Missing or incomplete experimental analyses, including sensitivity studies, GLO hyperparameter tuning, and broader empirical validation.

**Reviewer Scores:**

Reviewer PMRR: The reviewer is unlikely to change their rating given that major concerns remain unaddressed. Expected final score: 2.
(2 / 4)

Reviewer qwKR: Addressing the remaining concerns is unlikely to increase the score beyond the current rating. Expected final score: 6.
(6 / 4)

Reviewer SMmg: Several important concerns remain unresolved, making a score change unlikely. Expected final score: 4.
(4 / 3)

Reviewer 26a7: Similar to Reviewer SMmg, significant concerns remain unaddressed. Expected final score: 4.
(4 / 4)

The expected final average rating is 4.0 with 3.75 confidence.

---

### Decision · Program_Chairs · 2026-01-26

Reject